# Differential organization of tonic and chronic B cell antigen receptors in the plasma membrane

Maria Angela Gomes de Castro [1], Hanna Wildhagen[1], Shama Sograte-Idrissi[1,2], Christoffer Hitzing[3], Mascha Binder[4], Martin Trepel[4,5], Niklas Engels [3] & Felipe Opazo [1,2]

Stimulation of the B cell antigen receptor (BCR) triggers signaling pathways that promote the differentiation of B cells into plasma cells. Despite the pivotal function of BCR in B cell activation, the organization of the BCR on the surface of resting and antigen-activated B cells remains unclear. Here we show, using STED super-resolution microscopy, that IgM-containing BCRs exist predominantly as monomers and dimers in the plasma membrane of resting B cells, but form higher oligomeric clusters upon stimulation. By contrast, a chronic lymphocytic leukemia-derived BCR forms dimers and oligomers in the absence of a stimulus, but a single amino acid exchange reverts its organization to monomers in unstimulated B cells. Our super-resolution microscopy approach for quantitatively analyzing cell surface proteins may thus help reveal the nanoscale organization of immunoreceptors in various cell types.

[1] Institute of Neuro- and Sensory Physiology, University Medical Center Göttingen, Humboldtallee 23, 37073 Göttingen, Germany. [2] Center for Biostructural Imaging of Neurodegeneration (BIN), University of Göttingen Medical Center, von-Siebold-Straße 3a, 37075 Göttingen, Germany. [3] Institute of Cellular and Molecular Immunology, University Medical Center Göttingen, Humboldtallee 34, 37073 Göttingen, Germany. [4] Department of Oncology and Hematology, BMT with section Pneumology, University Medical Center Hamburg-Eppendorf, Martinistr. 52, 20246 Hamburg, Germany. [5] Department of Hematology and Oncology, Augsburg Medical Center, Stenglinstr. 2, 86156 Augsburg, Germany. Correspondence and requests for materials should be addressed to N.E. (email: nengels@gwdg.de) or to F.O. (email: fopazo@gwdg.de)

The characterization of the molecular arrangement of immunoreceptors on the cell surface has been hampered in the past by the lack of powerful imaging techniques that allow visualization and quantification of the entire pool of native receptor complexes within the plasma membrane in an unbiased manner. Thus, our knowledge on the structural organization of antigen receptors in lymphocytes is largely based on biochemical data and indirect visualization methods. Recent progress in the field of super-resolution microscopy now allows imaging and direct analysis of native receptors on the cell surface[1].

The conception of the molecular composition and spatial organization of the B cell antigen receptor (BCR) has changed considerably over time. Traditionally it was assumed that a fully assembled BCR complex adopts a symmetrical structure, in which one membrane-bound immunoglobulin (mIg) molecule makes non-covalent contacts to two copies of the signal-initiating Igα/Igβ (CD79A/B) heterodimer of transmembrane proteins[2–4]. Yet, when this model was put to the test it turned out that mIg and Igα/β are present in a 1:1 stoichiometry on the cell surface[5,6]. Another traditional assumption implied that BCR complexes consisting of mIg and Igα/β exist as 'monomeric units' on the cell surface of resting B cells. However, this view has been challenged in recent years by reports providing some clues that BCR units may form higher, oligomeric clusters in the plasma membrane of resting B cells, i.e., already in the absence of antigenic stimulation[7–9] These observations are based on experiments using indirect visualization methods like bimolecular fluorescence complementation (BiFC) or proximity ligation assay (PLA) aiming at determining the distance between individual BCR components (such as the mIg portion) or their capability to come into close proximity in the absence of antigen[7,8]. Furthermore, imaging experiments using direct stochastic optical reconstruction microscopy (dSTORM) indicated the existence of oligomeric BCRs containing several dozens of 'monomeric units' within so-called protein islands in the plasma membrane[9–11]. Based on these findings, it was proposed that the activation of intracellular signaling cascades following BCR stimulation requires the opening or dissociation of preformed BCR oligomers, which would expose the otherwise inaccessible immunoreceptor tyrosine-based activation motifs (ITAMs) within the cytoplasmic domains of Igα and Igβ to allow their phosphorylation by cytoplasmic protein tyrosine kinases (PTKs)[8,12]. This 'dissociation activation model' of BCR signal initiation basically reversed the traditional concept, according to which it is the antigen-induced clustering of predominantly monomeric BCR units that causes a local accumulation of otherwise scattered ITAMs to allow their efficient phosphorylation by PTKs[13–16].

However that may be, even in the absence of antigen the BCR seems to send signals into the cell that are essential for the survival of mature B cells in vivo[17–19]. This poorly defined survival or maintenance signal is believed to reflect an antigen-independent tonic activity of the BCR that may also involve a crosstalk with other cell surface proteins such as the BAFF receptor (also known as BR3) or Toll-like receptors[20,21].

In addition to this very low level of tonic maintenance signal, a constitutively elevated signaling activity of the BCR has been reported to be involved in survival and probably also formation of B cell-derived tumors, such as activated B cell-like diffuse large B cell lymphoma (ABC DLBCL) or chronic lymphocytic leukemia (CLL). Such chronically active BCR signaling can be brought about by mutations that cause amino-acid substitutions in the intracellular domains of Igβ or Igα in case of ABC DLBCL[22] or by auto-aggregation of BCRs in case of CLL[23–25]. CLL-derived Ig variable (V) domain sequences are remarkably stereotypic and have been shown to bind to self-epitopes in the V domains of neighboring BCRs[23,24,26–28]. A single amino-acid substitution within the self-epitope is sufficient to completely abolish the chronic signaling activity of CLL-derived BCRs[23]. Whether or not such chronic BCRs adopt a different organization in the plasma membrane than common, tonic BCRs with regard to clustering or oligomerization remains unknown.

Here we use stimulated emission depletion (STED) and dSTORM super-resolution microscopy techniques[29] to investigate the organization of native mIgM-containing BCRs with tonic and chronic signaling activity in human B cells. We observe that 'tonic' BCRs exist primarily as monomeric and dimeric units on the cell surface and form oligomeric clusters only when stimulated. In contrast, a CLL-derived 'chronic' BCR predominantly forms dimers and oligomers within the plasma membrane, which is reverted by a single amino acid substitution in its binding motif. In conclusion, our data reinforce the concept according to which antigen binding induces clustering of mostly monomeric BCR to initiate ITAM-controlled intracellular signaling reactions.

## Results

**A STED-based approach to determine the arrangement of BCRs.** To study the organization of native BCR complexes within the plasma membrane, it is essential to put the utmost attention to all technical details. Among all the most important details are the characteristics of the molecular probes used to detect the BCRs and the fixative procedure to immobilize the BCR complexes in the plasma membrane. Therefore, it was not only necessary to use a monovalent detection reagent to avoid artificial clustering of BCRs, but also to have a single fluorophore per affinity probe to enable a quantitative assessment of fluorescence intensities. These features can be found in a group of small recombinant proteins of ~7 kDa termed affibodies[30], which typically bind to their targets (here human Ig μ heavy chain) with high specificity and affinity. Importantly, each affibody molecule has only a single cysteine residue, which allows the site-directed conjugation of strictly one fluorescent dye molecule per molecule of affibody. We made use of this advantage and coupled the anti-human IgM affibody to the fluorophore Star635P (affibody-Star635P) to perform STED microscopy (for more details see Methods). To test the specificity of the fluorescently labeled affibody, we used Ramos B cells expressing endogenous mIgM-containing BCRs and a mIgM-negative Ramos sub-line (mIgM$^{neg}$) and stained both cell types with the affibody-Star635P. The fluorescently labeled affibody had a clear specificity for mIgM-expressing cells (Supplementary Fig. 1a, b). Importantly, the anti-IgM affibody-Star635P did not induce substantial activation of the BCR as assessed by intracellular Ca$^{2+}$ mobilization (Supplementary Fig. 1c). The subtle signal that we observed was most likely caused by traces of affibody dimers that can be formed during the fluorophore conjugation reaction via disulfide bonding of the single affibody cysteines. These dimers would be fluorophore-free and thus would not contribute to the fluorescence signal.

Since every conjugated affibody molecule was coupled to exactly one fluorophore molecule (degree of labeling was >97%), this affinity probe had a very homogeneous fluorescence intensity distribution that was determined by using STED microscopy on affibody-Star635P conjugates sparsely dispersed on glass coverslips (Supplementary Fig. 1e). We then tested whether we could use this affinity probe for quantitative imaging by incubating it with a recombinant monomeric human IgM-Fc molecule (which contains two Ig μ heavy chains) followed by STED microscopy as before (Supplementary Fig. 1e). The average intensity from single spots of monomeric IgM-Fc molecules incubated with Star635P-labeled affibodies was roughly 1.85 times brighter than that of single-fluorophore-labeled affibodies (Supplementary Fig. 1f).

This value is very close to the theoretical value of 2.0 times higher[31] and indicates that the majority of monomeric IgM-Fc molecules can indeed accommodate two affibody-Star635P molecules. In addition, we used microscale thermophoresis to determine the affibody:IgM-Fc stoichiometry as previously performed by Kazemier et al[32]. In agreement with our previous experiments, also the thermophoresis analysis indicates a 2:1 binding stoichiometry (affibody:IgM-Fc; Supplementary Fig. 1g, h).

Having established the specificity of the affibody against human IgM, its inability to activate the BCR and its suitability to quantitatively detect IgM-Fc monomers, we finally determined the saturating amount of affibody-Star635P that was needed to stain the cell surface pool of mIgM-containing BCRs on Ramos B cells (Supplementary Fig. 1i).

**Arrangement of mIgM-BCRs on the surface of human B cells.** We then used the exquisite (m)IgM-specific affibody affinity probe to investigate the organization and distribution of mIgM-containing BCRs on the cell surface of human B cells using STED microscopy. To this end we incubated live B cells with saturating amounts of fluorophore-conjugated affibody for 30 min on ice to minimize steady-state endocytosis of labeled BCRs[33]. After extensive washing, the cells were fixed using ice-cold buffer containing 4% paraformaldehyde (PFA) and 0.1% glutaraldehyde (GA) for 30 min. As previously demonstrated by Tanaka et al., incubation with a combination of PFA and GA is essential to effectively stop movement of membrane proteins[34]. Furthermore, ice-cold conditions prevent fixation-induced blebbing and swelling artifacts[35,36]. Labeled and fixed cells were allowed to sediment on glass coverslips and subsequently were exposed to a single sonication pulse. This treatment removed cell bodies, leaving on the coverslip intact plasma membrane patches that were in direct contact with the glass surface (Fig. 1a). This method to generate membrane sheets has been frequently used to study the distributions of membrane lipids and integral membrane proteins[37–41]. A major advantage of membrane sheets is that any fluorescence signal obtained by STED microscopy (or any other microscopy technique) selectively comes from the plasma membrane, thus minimizing signals from endocytosed BCRs. Moreover, this setting provided an excellent signal-to-noise ratio, which was required for observing single receptors. Using this protocol, we imaged dozens of membrane sheets from Ramos B cells and from primary human peripheral blood B cells obtained from healthy donors (Fig. 1b and e, respectively). Both primary B cells and Ramos B cells revealed the presence of discrete spots of mIgM-BCRs on their surfaces.

The fluorescence signal intensity distribution of the spots was then analyzed and compared to the intensity distribution of single fluorescent affibodies sparsely scattered on a coverslip, which was performed as the reference control for single-fluorophore intensities in every imaging session (Fig. 1c, f). An algorithm was created to find spots in a manually selected area of the membrane sheets (avoiding, for example, the borders of the membrane sheets where often double bilayer membranes can be found). Spots were first localized and their intensity line profiles on X and Y axes were fitted to a Gaussian distribution. Spots were considered 'true spots' only if the quality of their fits in X and Y was satisfactory and if their sizes (full width at half maximum, FWHM) were smaller than 700 nm. Due to the resolution limit of our STED instrument it was not possible to resolve any two spots closer than 50–60 nm. For further analysis we used the integrated fluorescence intensity (in the following "intensity" always means integrated intensity) of each spot to infer the number of fluorophores. The fluorescence intensity distributions of the analyzed spots on the membrane

sheets were fitted to a sum of Gaussian distributions for which the center position of each Gaussian curve was forced to be multiples of the center position determined by the single-fluorophore reference distribution (Supplementary Fig. 2a). From the amplitude of these Gaussian distribution fits, we derived the proportion of what we define as apparent monomeric, dimeric, and oligomeric mIgM-BCRs present in the B cell membrane sheets. Spots that contain one or two fluorescent affibody molecules are defined as apparent mIgM monomers, those that contain three or four affibody copies are defined as apparent mIgM dimers, and so forth (for illustration see hypothetical analysis in Supplementary Fig. 2b).

With this methodology and definition, roughly $56.7 \pm 2.9\%$ (mean ± standard deviation) of the BCRs in the plasma membrane of resting Ramos B cells represented apparent monomeric mIgM molecules. About $31.8 \pm 8.4\%$ of the mIgM-BCRs are consistent with dimers and only $10.9 \pm 8.9\%$ of them are found in small oligomers ($\geq 3$ mIgM-BCRs molecules per spot) (Fig. 1d, g and Supplementary Fig. 2a, c). Additionally, the remaining large clusters of BCRs that were not considered by the aforementioned 'true-spot' criteria were also considered by calculating their percentage signal intensities in relation to the total fluorescence signal derived from a membrane sheet. The signal contribution of these large clusters was calculated and represented $0.56 \pm 0.2\%$ of the total signal present in the membrane sheets (Fig. 1d, g). Noteworthy, the distribution trends of mIgM-BCRs were similar for both Ramos and primary human B cells. Please note that during the fluorophore conjugation reaction, smidgens of affibody molecules may have formed disulfide-bonded (non-fluorescent) dimers, which cannot efficiently be separated from the monomers due to their small molecular weight difference. These affibody dimers may have caused some otherwise monomeric mIgM molecules to arrange in higher ordered structures. However, the effect of such unintended clustering is probably very limited.

To test whether the distribution of BCRs changes upon their stimulation, we treated the cells with BCR-activating divalent F(ab')$_2$ fragments specific to human (m)IgM (see Supplementary Fig. 1c). As before, we incubated the cells on ice, since at higher temperatures activated BCRs are rapidly internalized[42–44], which interferes with our microscopic analysis of mIgM molecules in the plasma membrane. The kinetics of BCR-proximal signaling processes as tested by Ca$^{2+}$ mobilization are slightly decelerated under these conditions, but remain similar to those at more physiological temperatures (see Supplementary Fig. 1d). Hence, this experimental condition appears a suitable approach to stimulate BCRs while at the same time preventing their rapid endocytosis. The fluorescence intensities of mIgM-BCR-containing spots on Ramos B cells were greatly enhanced when the cells were exposed to this BCR-stimulating reagent, resulting in a higher proportion of oligomers and clustered mIgM-BCRs (Fig. 1h–j and Supplementary Fig. 2d). This observation is in agreement with the notion that clustering of mostly monomeric and dimeric BCRs is a crucial step in the activation of BCR signaling. At any rate, we did not observe a dissociation of preclustered oligomeric BCRs on addition of the stimulating reagent.

In addition, we tested whether our methodology can be used to investigate the organization of mIgM-BCRs in native B cells using monovalent Fab fragments, which are usually readily available for most cell surface receptors. To this end we repeated the experiments shown above with a Fab against human (m)IgM coupled to the Star635P fluorophore (Supplementary Fig. 3a–d). As before, membrane sheets of Ramos B cells stained with the anti-IgM Fab-Star635P probe displayed discrete spots and fluorescence intensity distributions similar to the ones obtained with the affibody-based affinity probe (Supplementary Fig. 3e).

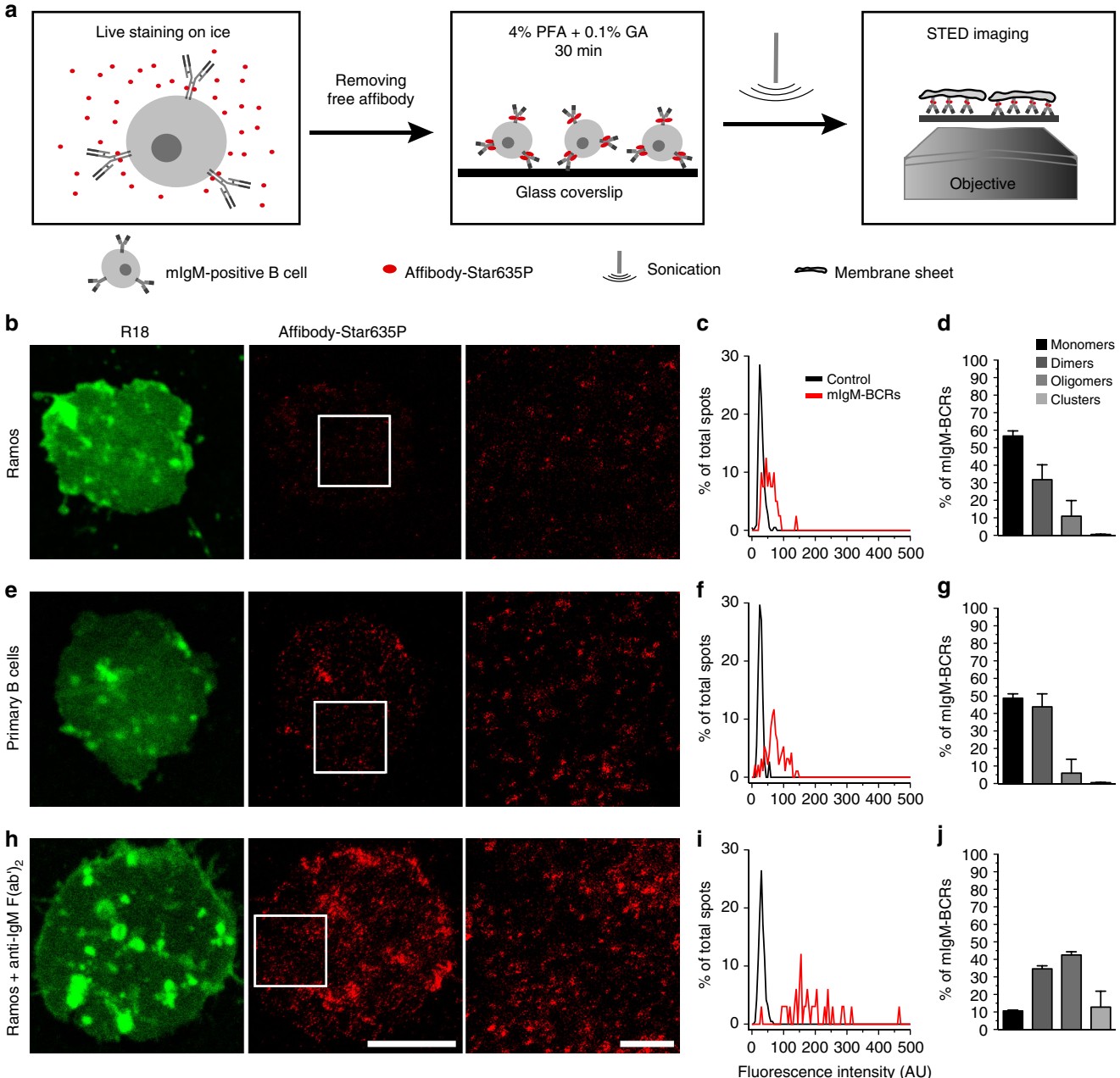

**Fig. 1** Arrangement of mIgM-containing BCRs in the plasma membrane of human B cells. **a** Schematic depiction of the staining procedure and the generation of plasma membrane sheets. **b**, **e**, **h** Images of membrane sheets of unstimulated Ramos B cells (**b**), unstimulated primary B cells (**e**), and Ramos cells stimulated with anti-IgM F(ab')$_2$ fragments for 30 min on ice (**h**). Membrane sheets were stained with the membrane marker R18 (confocal images pseudo-colored in green) and anti-human IgM affibody-Star635P (STED images displayed in red). White squares on central images indicate the zoomed regions that are displayed in the images to their right. Scale bars represent 5 μm and 1 μm for the low and high zoom images respectively. **c**, **f**, **i** Example histograms showing the fluorescence intensity distributions of single Star635P-conjugated affibodies on coverslips (black curves, control) and mIgM-spots present in membrane sheets (red curves, mIgM-BCRs). **d**, **g**, **j** Percentages of different mIgM-BCR arrangements in plasma membrane sheets were calculated after pooling the spot intensities of dozens of membrane sheets together and then fitting the intensity distribution to a sum of Gaussian curves (Supplementary Fig. 2). Data were obtained from four and three independent experiments including a total of 30 membranes from unstimulated and 15 membranes from F(ab')$_2$-stimulated Ramos B cells, respectively. The experiments with primary human B cells were performed three independent times from two different donors and 30 membrane sheets were analyzed in total. Error bars represent the 68% confidence interval, corresponding to 1 standard deviation. Histogram's source data are provided as a Source Data file

Moreover, when the cells were activated with divalent F(ab')$_2$ fragments against human IgM, large clusters appeared together with the characteristic shift in the fluorescence intensity distribution pattern (Supplementary Fig. 3f). Unfortunately, the Fab-Star635P was not suitable to determine the organization of mIgM-BCRs in a quantitative manner. This was mainly due to

the unknown number of fluorophore molecules per Fab molecule and the broad fluorescence intensity distribution profiles when analyzing single monomeric immunoglobulin μ heavy chains by STED microscopy. Nonetheless, also with this staining strategy we were not able to detect preformed oligomeric clusters of mIgM-BCRs on the surface of resting B cells.

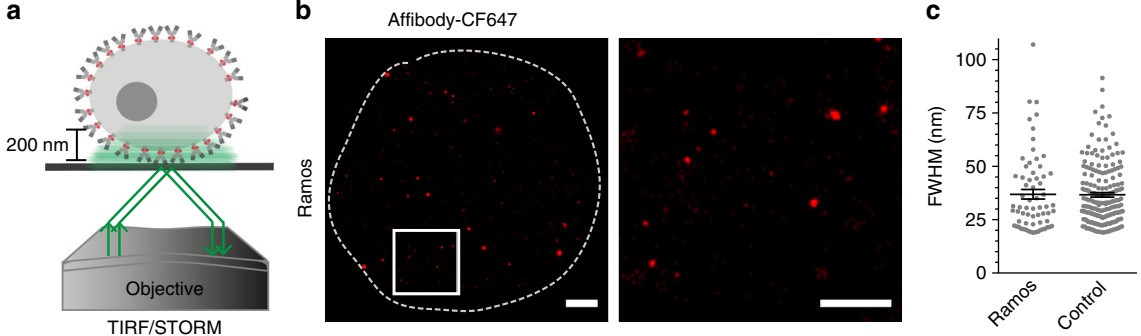

**Fig. 2** mIgM-BCRs do not arrange in large protein islands at the plasma membrane. **a** Schematic depiction of the TIRF/dSTORM setup imaging depth. **b** Representative TIRF/dSTORM image of an intact Ramos B cell stained with CF$^{TM}$647-conjugated anti-IgM affibodies. The dotted white trace delineates the area of the plasma membrane close to the glass coverslip imaged under TIRF/dSTORM mode. The white square indicates the zoomed area shown in the image to the right. Scale bars represent 1 μm and 0.5 μm for the low and high zoom, respectively. **c** Scatter plot displaying the spot size (FWHM) distribution unstimulated Ramos cells and as resolution reference the scatter plot of single CF$^{TM}$647- affibodies spread on a glass coverslip

**Unstimulated mIgM-BCRs do not assemble in large islands.** Previous studies using dSTORM reported the existence of large clusters (or islands) of mIgM-containing BCRs on the surface of resting B cells with average diameters of either ~150 nm[9] or several hundred nm[10], which would accommodate an average of roughly 30–50 mIgM-BCRs[9,10]. To test whether this discrepancy to our findings was due to the different imaging techniques used, we employed dSTORM imaging to study the organization of mIgM-BCRs on the surface of intact cells. For this purpose, we stained Ramos B cells with the anti-IgM affibody, this time coupled to CF$^{TM}$647 fluorophores (affibody-CF647, DOL 0.97, Supplementary Fig. 4). dSTORM imaging could be performed with intact cells due to the use of a total internal reflection fluorescence (TIRF) setup (Fig. 2a). The TIRF/dSTORM configuration allowed us to selectively image the proximal ~200 nm into the sample from the glass coverslip[45–47], which is several folds better than the ~700 nm axial resolution of a confocal microscope. This enabled us to super-resolve mIgM-BCRs in the plasma membrane of intact B cells. After staining the cells with saturated concentrations of fluorescent affibody (Supplementary Fig. 4), the average FWHM of the spots present in the plasma membrane was below 37 nm. However, the median of 31 nm suggests that most of the spot sizes were actually smaller than the average and very similar to the maximal average resolution of 36 nm achieved by imaging single CF647-labeled affibodies on a glass surface (Fig. 2c). Importantly, in our samples that were fixed with both PFA and GA, we detected only very few large clusters (or islands) that were greater than 70 nm in diameter after staining with fluorescent affibody. Although it is very difficult to be quantitative in dSTORM imaging due to several potential artifacts that can lead to inaccurate results[48–53], our observations consistently suggest that mIgM-BCRs on the surface of unstimulated B cells can be found in different molecular arrangements. However, the majority of cell surface BCRs seems to be arranged as monomers and dimers, but not in preformed large islands as previously suggested.

**Dimeric and oligomeric organization of a CLL-derived BCR.** Since the constitutive (or chronic) signaling status of BCRs has been linked to tumor formation especially in CLL, we aimed to employ our imaging strategy to investigate the molecular arrangement of a CLL-derived BCR on the plasma membrane of human B cells. Specifically, we wanted to address whether CLL-derived BCR complexes show signs of auto-aggregation on the cell surface, since the V region sequences of CLL-derived Ig heavy chains have been shown to mediate autonomous chronic

activation of CLL-BCRs, most likely by interacting with epitopes within the Ig heavy chain V (VH) regions of neighboring BCRs on the same cell[23,24].

To analyze the clustering status of CLL-derived BCRs, we generated a cellular model system that allowed us to study a CLL-BCR in a genetically tractable manner and a defined cellular context. To this end we cloned the V gene segment of the Ig light (VL) locus from an unmutated CLL (U-CLL, for details see Methods) sample and expressed it constitutively in a variant of the human Burkitt lymphoma cell line DG75, which has lost expression of its endogenous mIgM-BCR (mIgM$^{neg}$). In addition, we cloned the corresponding VH region from the same CLL sample and expressed it in a doxycycline-inducible manner in the context of a membrane-bound Ig μ heavy chain. To test our expression system, we first analyzed the levels of 'CLL-derived' mIgM on the cell surface by flow cytometry (Supplementary Fig. 5). Super-resolution STED microscopy was then used to image membrane sheets of these cells as well as the parental DG75 B cell line, which expresses an endogenous mIgM-BCR (of unknown specificity). Analysis of the endogenous BCR of DG75 cells showed a very similar fluorescence intensity distribution profile as observed in Ramos and primary B cells (see Fig. 1) with 67.6 ± 3% of the BCRs being apparent monomers and 29.8 ± 8.5% corresponding to apparent dimers of mIgMs (Fig. 3a–c). Strikingly, this ratio was inverted in DG75 cells expressing the CLL-derived BCR (Fig. 3d–f), indicating that the CLL-derived Ig V regions indeed caused auto-dimerization and oligomerization of mIgM-BCRs on the cell surface. Additionally, a slight increase in the number of larger clusters (4.01 ± 0.5%) was also observed in cells expressing the CLL-derived BCRs (Fig. 3f).

To test whether the differences in the apparent dimer formation between the tonic BCRs from primary and Burkitt lymphoma B cells and the CLL-derived BCR are indeed caused by autonomous self-aggregation of the latter, we replaced an arginine residue within the CLL VH domain, which lies within a motif that was reported[23] to serve as 'auto-antigen' for chronic activation of CLL-derived BCRs, with alanine (R38A) and analyzed this mutant CLL-BCR as before. Indeed, this single amino acid substitution fully reversed the auto-aggregation displayed by the CLL-derived BCR and converted it into a mostly monomeric receptor (Fig. 3g–i), showing that self-reactivity of this CLL-BCR caused it to arrange as dimers and oligomers.

## Discussion
Here we have shown that tonic mIgM-containing BCRs of primary human B cells are apparently organized as monomers and

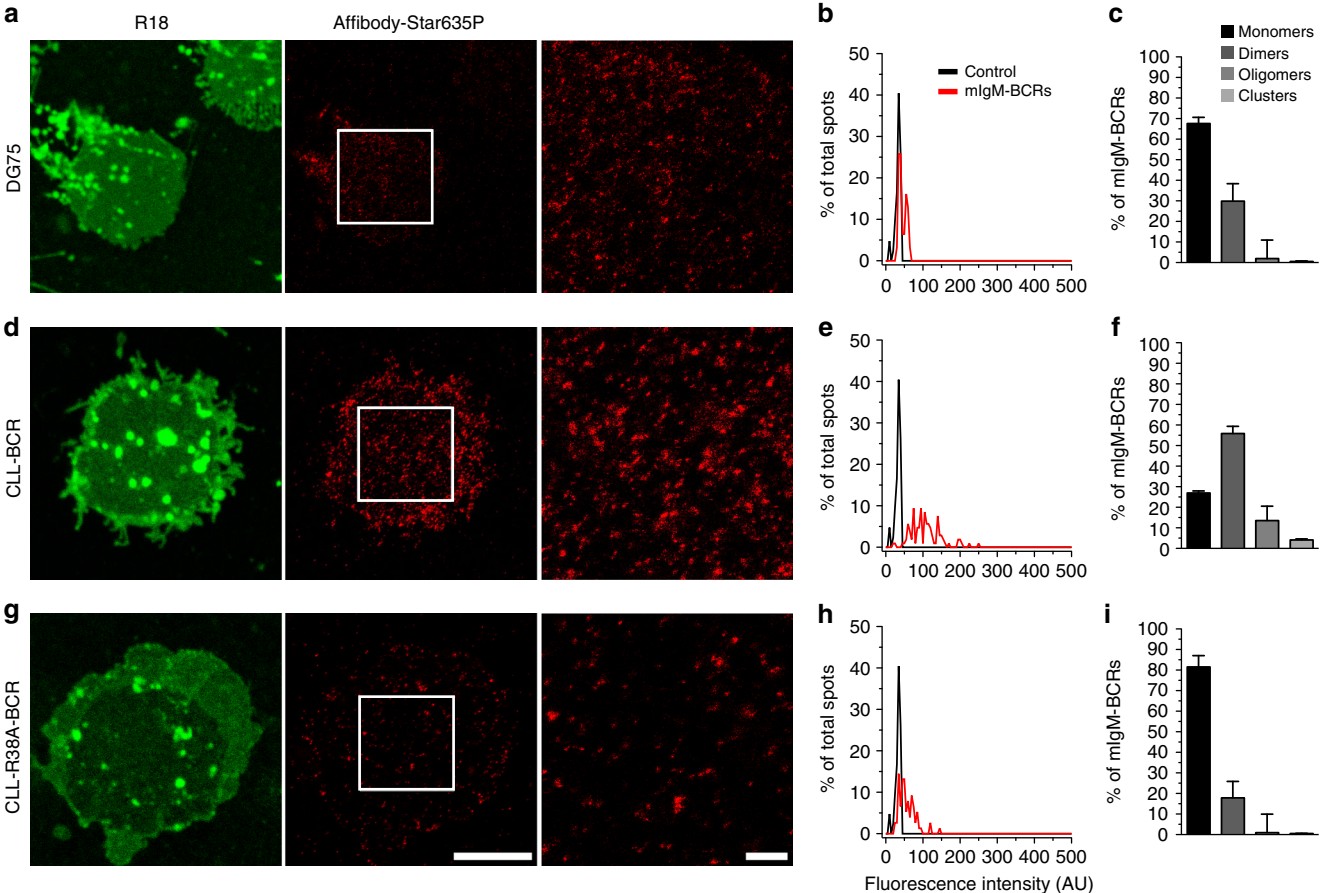

**Fig. 3** Cell surface arrangement of a CLL-derived mIgM-BCR. **a, d, g** Plasma membrane sheets of DG75 B cells (**a**), an mIgM[neg] variant expressing a CLL-derived mIgM-BCR (CLL-BCR, **d**) or the R38A-mutant variant thereof (CLL-R38A-BCR, **g**) were analyzed as in Fig. 1. White squares indicate the zoomed regions. Scale bars represent 5 µm for the low zoom images and 1 µm for the high zoom STED images. **b, e, h** Example histograms of the spot fluorescence intensities for each cell type. **c, f, i** Percentages of different mIgM-BCR arrangements were derived as in Fig. 1. Data were obtained from three independent experiments including 16 membranes for DG75, 17 membranes for the CLL-BCR, and 12 membranes for the CLL-R38A-BCR (raw data in Supplementary Fig. 5). Error bars represent the 68% confidence interval, corresponding to 1 standard deviation. Histogram's source data are provided as a Source Data file

dimers within the plasma membrane of resting cells, whereas a CLL-derived chronic BCR tends to be organized as dimers and oligomers, which is dependent on the presence of an intact recognition motif within the Ig heavy chain V region of the CLL-derived BCR. Previous reports using biochemical and indirect visualization approaches suggested that BCRs on the surface of resting cells might accommodate larger, oligomeric clusters, which were hypothesized to adopt a closed, probably auto-inhibiting conformation[5,7,8,12]. Also, independent studies using dSTORM microscopy imaging approaches indicated the existence of antigen-independent BCR oligomers on the B cell surface, even though the results of these studies differed considerably[9–11]. With our STED-based imaging method we did not detect such an oligomeric organization of resting BCRs. This was not due to technical limitations of our approach, since we were able to detect large mIgM-BCR clusters following stimulation with a BCR-activating divalent reagent. However, also our fluorescence intensity-based method has some intrinsic constraints. For instance, our definition of apparent BCR monomers, dimers, and oligomers may be complicated by a potential non-saturated binding of the affibody to (m)IgM molecules. Thus, our results tend to have a slight bias towards smaller cluster sizes, since e.g., a spot containing two fluorophores would in our analysis be classified as monomer, but theoretically could also be a dimer with extremely poor (only 50%) labeling efficiency. Furthermore, the

fluorescence intensity distribution profiles of fluorescent spots typically get broader with increasing brightness, which limits the accuracy of our method to estimate the number of BCRs in spots that are equivalent to approximately six or more fluorophores. Nevertheless, we did not observe a large proportion of bright spots (≥6 fluorophores) on resting B cells. In fact, the vast majority of the spots we found in the plasma membranes of resting cells displayed fluorescence intensity distributions consistent with monomeric and dimeric organization of BCRs.

Our novel imaging approach allowed us to visualize relatively subtle changes in BCR clustering that were caused by auto-aggregation of a chronic CLL-derived BCR in the absence of an exogenous BCR crosslinker. Hence, our method can detect (oligomeric) BCR clusters if they exist. This raises the question as to why other groups have come to different results. One of the most critical parameters that has to be treated with the utmost care is the immobilization of membrane proteins during the fixation step to avoid any chance of lateral movement following staining of the receptors. It has been demonstrated that poor fixation (10 or even 30 min using 4% paraformaldehyde) will not stop membrane proteins from lateral difusion[34]. Therefore, lateral diffusion of insufficiently fixed BCRs poses a major problem in the interpretation of dSTORM images[10,11,34]. Our fixation procedure, which included a combination of ice-cold paraformaldehyde and glutaraldehyde, was chosen to make sure that transmembrane

proteins are clearly immobilized before we prepared membrane sheets for microscopic analyses[34–36].

Furthermore—unlike indirect visualization strategies—our approach is not biased since it does not require BCRs to be in close proximity to be detected as is the case for the PLA or the BiFC method, which both fail by design to detect monomeric components and thus inevitably mask out the presence of monomeric BCRs[8,10]. In line with our observations, a previous study using Förster resonance energy transfer (FRET) experiments could not provide evidence for the existence of BCR oligomers in resting cells[6]. Interestingly, also the T cell antigen receptor (TCR) was recently shown by a combination of different super-resolution imaging approaches to exist predominantly as randomly distributed monomeric units on the T cell surface[54,55].

The STED microscopy technique takes advantage of controlling the transitions between on and off states of a fluorophore in a defined space (pixel-by-pixel) while acquiring the image. This results in a direct super-resolution image without the need of complex data processing as is the case for localization techniques like dSTORM[56]. Importantly, there are several known potential artifacts that can be generated when imaging with single molecular localization techniques. Some of such artifacts can be caused by high density labeling, low irradiation power, and overcounting localizations, all of which can strongly influence the size and structure of the objects under investigation[48–53]. Another major factor that allowed us to estimate the membrane organization of BCRs was the use of affibodies. Small and monovalent affinity probes like nanobodies, aptamers, or affibodies are preferred as accurate tools for current super-resolution microscopy techniques[29,57,58]. Here we have used an affibody against human IgM, which is only ~7 kDa and ~2 nm in size and thus is superior to e.g., antibodies in accessing epitopes. The monovalent binding property of fluorophore-conjugated affibodies having only traces of non-fluorescent dimers minimizes the risk of inducing artificial clustering of target molecules, allowing its incubation with live cells prior to fixation. Another unique feature of affibodies, which makes them ideal for quantitative imaging, is that they can be conjugated with exactly one flurophore molecule per affibody in a site-directed manner. We usually obtained between 0.97 and 0.99 moles of fluorophores per mol of affibody, which allowed stoichiometric detection of endogenous levels of BCRs in cellular membranes. This is typically not possible with other affinity probes like immunoglobulins or even Fab fragments that are randomly conjugated to fluorophores. Despite the small size of affibodies we cannot rule out that steric hindrance may have caused incomplete labeling of large BCR clusters. Thus, even this labelling strategy may not be ideally suited to determine the exact number of receptors in large clusters.

Another factor that complicates the analysis of our spot intensity histograms is that the mean and the width of the single-fluorophore-intensity distribution (SFID) are on the same order of magnitude, and that the width of the peaks increases with the number of fluorophores per cluster ($\sqrt{n}$, see formula in Methods). This leads to histograms with overlapping peaks that cannot clearly be resolved by eye (see Suppl. Figures 2 and 5). Please note that the width of the SFID has intrinsic reasons such as e.g., the random distribution of the fluorophores' dipole orientations. However, despite the overlapping peaks, we can still extract robust information on the composition of cluster sizes from the histograms by using the knowledge on the mean and the width of the SFID, which we obtained from the single-fluorophore calibration measurements (Suppl. Figure 2a). Thus, only the amplitudes of the peaks ($a_n$, see formula in Methods) are left to be extracted from the fits, thereby yielding robust results (indicated by e.g., reasonably large $R^2$ values and small confidence intervals). Nevertheless, we observed that the model did not always perfectly

fit the experimental data (see Suppl. Figures 2 and 5). While part of the deviations can be attributed to Poissonian number fluctuations, some deviations do not look purely statistical. These deviations might be caused by instrument drift during imaging sessions and/or the microenvironment of the fluorophores being slightly different in the plasma membrane as compared to the conditions of the calibration measurements. Despite these small imperfections, the fit results are robust enough to support our conclusions.

Stimulation of tonic BCRs with an activating reagent induced formation of large clusters as anticipated by the crosslinking model of BCR activation[16]. This observation does not concur with recently proposed models according to which BCR stimulation induces the dissociation of pre-clustered oligomeric BCRs to initiate intracellular signaling. Instead, our results reinforce the concept that antigen binding to many monomeric and dimeric 'BCR units' causes aggregation of ITAMs that in the absence of antigen are dispersed in the plasma membrane. In this scenario antigen functions as a classical adaptor molecule that connects signaling components (here BCR complexes) that otherwise come together only in a very inefficient manner. This crosslinking of BCRs might stabilize lipid-ordered domains in the plasma membrane that facilitate BCR activation[14,59]. However, a considerable pool of tonic mIgM-BCR spots contains apparent receptor dimers and a small percentage seems to contain oligomers already in the absence of antigen. The mechanisms that cause dimerization of a large fraction of tonic surface mIgM-BCRs remain to be investigated. However, it has been demonstrated that the actin cytoskeleton controls the lateral movement of BCRs by constituting barriers by which receptors are immobilized[14,15,20]. Hence, dimers of tonic mIgM-BCRs may reflect two monomeric BCRs that are trapped within a membrane skeleton network compartment[40,60], rather than being held together by an intrinsic affinity for each other. Nevertheless, it is tempting to speculate that the dimeric or oligomeric pool of BCRs provides the tonic maintenance signal of the BCR that has been demonstrated in several mouse models to be critical for survival of mature B cells[17–19,61]. If indeed the dimerization of tonic BCRs is sufficient to provide B cells with a survival signal, a relatively moderate increase in the amount of BCR dimers and oligomers that we observed for a CLL-derived BCR might be enough to tip the balance toward chronic, pathological signaling.

We believe that our newly established quantitative STED-based super-resolution microscopy approach will be a valuable method to study the nanoscale organization of various immunoreceptors not only in B cells and thus will be very useful to provide a better understanding of the surface 'landscape' of immune cells and the mechanisms that underlie immune receptor function.

## Methods
**Cells and expression vectors**. The human Burkitt lymphoma B cell lines Ramos and DG75 were purchased from the German Collection of Microorganisms and Cell Cultures (DSMZ, Braunschweig, Germany). DG75EB cells expressing the murine cationic amino-acid transporter 1, which makes them susceptible to infection with MMLV-based retrovirus particles were described before[62] and were used in all experiments. Ramos and DG75EB cells lacking expression of endogenous mIgM were obtained by repetitive sorting of mIgM-negative cells identified by staining with a polyclonal anti-human IgM antibody (SouthernBiotech, Birmingham, USA). IgM-negative DG75EB cells were retrovirally transfected to express a human λ Ig light chain derived from a patient CLL sample (#025). These cells were additionally transfected with the doxycycline-controllable expression vector pRetroX-TetOne™-Puro (Clontech Laboratories Inc., Montain View, USA), encoding a human membrane-bound Ig μ (μm) heavy chain containing the variable heavy (VH) region from CLL sample #025. The CLL #025 VH region is composed of *IGHV1-69*, *IGHD3-22*, and *IGHJ6*, all of which are in germline configuration. The resulting CDR3 amino-acid sequence is CARFMDYYDSS-GYYNPQYYYYGMDV (single letter code for amino acids). The arginine residue at position 38 of the CLL#025 VH domain was replaced with alanine (R38A) by site-directed mutagenesis. Primers used for this cloning and mutagenesis are in

Supplementary Table 1. Expression of the CLL-derived μm heavy chain variants was induced by addition of 5 μM doxycycline to the cells 18–24 h before staining. All cell lines were cultured at 37 °C in a humid atmosphere with 5% CO$_2$ in RPMI 1640 medium, (Gibco/Life Technologies, Carlsbad, USA or Biochrom, Berlin, Germany), supplemented with 10% FBS (Biochrom, Berlin, Germany), 4 mM L-glutamine and 100 U per ml each penicillin and streptomycin (Lonza, Basel, Switzerland). Primary human B cells were isolated from peripheral blood of healthy donors by a negative magnetic cell sorting strategy using the 'B cell isolation kit II' (Myltenyi Biotec, Bergisch Gladbach, Germany) according to the manufacturer's instructions. The purity of the isolated B cells was assessed by staining of CD19 and mIgM and was usually greater than 90%. Experiments involving human participants were approved by the ethical review committee of the University Medical Center Göttingen and were performed in accordance with relevant guidelines and regulations. An informed consent was obtained from all participants.

**Flow cytometry and measurement of intracellular free Ca$^{2+}$.** Cell surface expression of mIgM was analyzed by flow cytometry using Alexa647-labeled polyclonal goat anti-human IgM antibodies (SouthernBiotech, Birmingham, Alabama, USA). Data were acquired with a FACS Calibur or LSRII (BD Biosciences). Analysis was done using FlowJo software (FlowJo, LLC, Ashland Oregon, USA). For analyzing intracellular Ca$^{2+}$ mobilization kinetics, $1 \times 10^6$ cells per measurement were incubated in RPMI 1640 containing 10% FCS, 1 μM Indo-1-AM (Molecular Probes), and 0.015% Pluronic-F-127 for 30 min at 30 °C. The cells were then washed and suspended in Krebs Ringer solution (10 mM HEPES (pH 7.0), 140 mM NaCl, 4 mM KCl, 1 mM MgCl$_2$, 1 mM CaCl$_2$, and 10 mM glucose). Basal Ca$^{2+}$ levels were measured for 30 s after which the cells were stimulated with the indicated reagents. Measurements were done with a LSRII flow cytometer (BD Biosciences) and data analysis was done using FlowJo software (FlowJo, LLC) and Microsoft Excel. A forward scatter/side scatter gate was used in all flow cytometric analyses to identify live cells and exclude cellular debris.

**Conjugation of affibodies or Fab to fluorophores.** Unconjugated anti-IgM affibody® ab36088 (Abcam, Cambridge, UK) containing a single C-terminal cysteine was coupled either to the maleimide variant of Star635P (Aberrior, Göttingen, Germany) or to CF$^{TM}$647 (Biotium, Fremont, USA) for STED and dSTORM microscopy, respectively. Approximately 70 nmol of affibody was incubated for 60 min at room temperature in 500 μl of reducing buffer (50 mM HEPES, 150 mM NaCl, 10 mM EDTA, and 100 mM TCEP at pH 7.0). The reducing buffer including TCEP was exchanged using a NAP5 desalting column (GE Healthcare), leaving the reduced affibody in argon-degassed 50 mM HEPES, 150 mM NaCl, 10 mM EDTA at pH 7.6. The affibody was immediately mixed with ~15 molar excess of the maleimide fluorophore to ensure that most if not all affibodies were conjugated to a fluorophore. The coupling reaction was incubated at room temperature for 60 min. Finally, cysteamine chloride was added (50 mM) for 15 min at room temperature to quench remaining reactive fluorophores. Fluorophore-conjugated affibodies were separated from free fluorophores by using a self-packed desalting column with G-25 superfine media (GE Healthcare, USA). The resulting degree of labeling (DOL) was 0.99 mol of dye per mol of affibody for the Star635P and 0.97 for the CF$^{TM}$647. The stock of Star635P-maleimide or CF$^{TM}$647-maleimide were prepared in anhydrous DMSO at a final concentration of 10 μg per μl. Unconjugated Fab fragment (0.5 mg) was brought to pH ~8.5 with 0.1 M NaHCO$_3$ and incubated with ~8 molar excess of NHS-ester-Star635P for 60 min at room temperature under constant agitation and protected from light. Remaining reactive dye was quenched by adding excess of 1,5 M hydroxylamine at pH 8.5 and constant agitation for 30 more min. Finally, fluorophore-conjugated Fab fragments were separated from free fluorophores by using a self-packed desalting column with G-25 sephadex superfine media (GE Healthcare, USA). To determine the degree of labeling (DOL), we first determine the protein concentration in Molarity (M) as follow:

$$\text{Prot. conc.}(M) = \frac{A_{280} - (A_{max} \times CF)}{\varepsilon} \qquad (1)$$

Where $A_{280}$ is the absorbance at 280 nm, $A_{max}$ is the absorbance value of the dye at its maximal absorbance wavelength for the particular dye molecule. Correction factor (CF) is a constant specific for each dye molecules to compensate the amount of absorbance at 280 nm caused by the dye (information obtained from the dye supplier). Finally, $\varepsilon$ is the protein molar extinction coefficient in M$^{-1}$ cm$^{-1}$. Finally, to determine the DOL we need the molar extinction coefficient of the fluorescent dye $\varepsilon'$ (information obtained by the dye supplier) and the following equation.

$$\text{DOL} = \frac{A_{max} \text{ (of labeled protein)}}{\varepsilon' \times \text{Prot. conc.}(M)} \qquad (2)$$

**Staining of B cells with affibodies or Fab.** B cells (~200,000 per sample) were centrifuged in 1.5 ml tubes for 4 min at 300 x $g$, and cell pellets were resuspended with 50 μl ice-cold complete medium supplemented with fluorescently labeled anti-IgM affibody or fluorescently labeled Fab fragment. After incubation on ice for 30 min, cells were washed three times in ice-cold Dulbecco's PBS (DPBS). After the

last washing step, the pellets were resuspended in 1 ml of ice-cold fixation solution composed of 4% paraformaldehyde and 0.1% glutaraldehyde in DPBS and transferred to one well of a 12-well plate containing poly-L-lysine- (PLL) coated coverslips. The plate was centrifuged at ~100 x $g$ for 10 min at 4 °C to allow the cells to adhere to the coverslips. Plates were then incubated at room temperature in the dark for 30 min. The fixation solution was removed and 1 ml of quenching solution composed of 0.1 M glycine in DPBS was added to each well. After 15 min of quenching, cells were washed three times with DPBS. For whole cell imaging (e.g., Figure 3 and Supp. Figure 1 and 2) fixed cells were additionally stained for 10 min with 1 μg per ml of the membrane marker R18 (octadecyl rhodamine B chloride from Molecular Probes$^{TM}$, Oregon, USA). After washing once with DPBS, coverslips were either embedded for STED in Mowiol mounting media (6 g glycerol, 6 ml deionized water, 12 ml 0.2 M Tris buffer pH 8.5, 2.4 g Mowiol® 4–88, Merck Millipore). For dSTORM imaging, coverslips were first incubated with fluorescent beads (FluoSpheres®carboxylate-modified, 0.02 μm, crimson (625/645), diluted 1:1,000,000 in 0.1 mg ml$^{-1}$ PLL) for drift correction. Afterwards, coverslips were mounted using microscope slides with a cavity of ~15–18 mm in diameter and 0.6–0.8 mm in depth (Marienfeld Superior, Lauda-Königshofen, Germany, Cat No. 1320002) in order to have the samples embedded in the buffer required for dSTORM imaging. This buffer has a pH of 8.0 and is composed of 50 mM Tris/HCl, 10 mM NaCl, 10 mM β-mercaptoethylamine (MEA), 10% glucose, 2000 U ml$^{-1}$ catalase, 50 U ml$^{-1}$ glucose oxidase, similar to what has been described in earlier dSTORM studies[63,64].

**Generation of membrane sheets.** Stained and fixed cells (as described above) were exposed to a brief and a single ultrasound pulse using a Sonopuls sonifier HD2070 (Bandelin Electronic, Berlin, Germany). Coverslips containing the cells were placed in the center of a glass container dish (Ø 95 mm, Duran®, Carl Roth, Germany) covered with ~150 ml of ice-cold sonication buffer (0.1 M glycin in DPBS). The sonicator tip was positioned in the buffer at a distance of ~1 cm from the coverslip and a one second ultrasound pulse with 10% power was applied. After the sonication pulse coverslips were transferred to a 12-well plate containing DPBS supplemented with 1 μg ml$^{-1}$ of the membrane dye R18 (Molecular Probes, Oregon, USA). After washing twice in DPBS, samples were embedded in Mowiol and slides were stored at 4 °C protected from light until imaging the next day.

**Induction of mIgM clustering.** Live cells were stained on ice for 30 min with fluorophore-labeled affibodies or Fab fragments as described above followed by addition of 10 μg ml$^{-1}$ of polyclonal goat anti-human IgM, Fc5μ fragment-specific F(ab')$_2$ (Jackson ImmunoResearch, West Grove, PA, USA) or mouse anti-human IgM mu chain monoclonal antibody ab99374 (1:100) (Abcam, Cambridge, UK) for another 30 min on ice. All subsequent steps including washing, fixation, quenching, preparation of membrane sheets, and staining with R18 were done as described above.

**Reference signal from affibodies or Fab fragment.** Fluorophore-conjugated affibodies or Fab fragments were diluted 1:5000 in DPBS supplemented with 1% bovine serum albumin (BSA) and seeded on poly-L-lysin- (PLL) treated glass coverslips. After incubation for 1 h, coverslips were briefly washed in a large volume of filtered DPBS (0.2 μm filter size) and finally mounted in Mowiol. These samples were prepared and imaged in every imaging session and their spot fluorescence intensity distributions were used to calibrate the single-fluorophore intensities. Similarly, to determine the number of affibodies that bound on average to a single monomeric IgM molecule, 40 pmol of recombinant monomeric IgM-Fc molecules (Absolute Antibody, catalogue # Pr00108-15.5) was incubated in solution for 1 h at room temperature with 80 pmol of labeled anti-IgM affibodies to allow binding. To remove unbound labeled affibodies, the mixture was further incubated with 100 μl of high-capacity streptavidin-agarose beads (GE Healthcare) preloaded with biotin-conjugated human pentameric IgM (Rockland Inc., Pennsylvania, catalogue # 009-0607) for 30 min at room temperature. After a short centrifugation, the supernatant was incubated with a PLL-coated glass coverslip for 1 h, washed, mounted, and imaged as described above. To determine the number of Fab fragments that bound on average to a single monomeric μ heavy chain molecule, 1 nmol of purified μ chain (American Research Products Inc., catalogue # 13-3035-2) was incubated in solution with 2–4 nmol of labeled Fab fragment for 1 h at RT. Complexes were then seeded on PLL-coated glass coverslips and incubated for 1 h at room temperature. After a washing step using a large volume of filtered DPBS, coverslips were mounted in Mowiol.

**Microscale Thermophoresis.** The anti-IgM affibody K$_d$ was measured with a NanoTemper Microscale Thermophoresis Monolith NT.115Pico (NanoTemper Technologies GmbH, Munich, Germany). Five nsnomolar of affibody-Star635P was incubated with different concentrations (ranging from 0.05 to 1750 nM) of recombinant IgM-Fc (Absolute Antibody, catalogue # Pr00108-15.5). All dilutions were made using the MST buffer supplied together with the premium capillary by the manufacturer. The curve fitting and the K$_d$ calculation were performed using non-linear regression for binding saturation from GraphPad Prism. To determine the stoichiometry of the interaction between IgM-Fc and affibody a mixture of 5 nM of affibody-635P and 395 nM of unlabeled affibody was used to achieve

saturating conditions, but not to saturate the detector of the microscale thermophoresis device. The thermophoresis saturation point was determined using concentrations of recombinant IgM-Fc ranging from 18 to 1400 nM (16 points with 3:1 dilution). Linear regressions of the saturated and non-saturated data points were performed as previously described by Kazemier et al[32].

**Imaging**. A conventional epifluorescence Olympus IX71 microscope equipped with 0.75 NA/60x oil objective and an Olympus F-View II CCD camera (Olympus, Hamburg, Germany) was used for determining affibody specificity and saturating staining conditions (Supp. Figure 1,3, and 4). Confocal and STED images were obtained with a True Confocal System STED SP5 fluorescence microscope (Leica Microsystems, Mannheim, Germany) equipped with a 100 × 1.4 NA PL APO CS oil objective (Leica Microsystems, Mannheim, Germany). For confocal imaging of membrane sheets stained with R18 (Figs 1 and 3 and Supplementary Fig. 3), 1024 × 1024 pixels images with a pixel size set to 50.5 and 20.2 nm, respectively, were scanned at 1 kHz speed. Confocal signals were detected using a photomultiplier after Airy 1 pinhole size. Excitation of rhodamine dye (R18) was performed with a 543 nm helium-neon laser line and the signal was detected between 565 and 622 nm. Excitation of Star635P was done with a 633-nm helium-neon laser and the signal was detected between 647 and 730 nm. For STED imaging of (m)IgM molecules on membrane sheets and reference coverslips (Figs. 1, 3 and Supplementary Fig 3), pixel size and scanning speed were set to 20.2 nm and 1 kHz, respectively, and the signal was detected with an avalanche photodiode detector (APD). Excitation was performed with a 640 nm diode laser and depletion with a MaiTai pulsed tunable laser at 750 nm (Mai Tai Broadband, Spectra-Physics, Santa Clara, CA, USA). dSTORM images were obtained with an Olympus IX83 fluorescence microscope equipped with a x100 total internal reflection fluorescence (TIRF) objective. CF$^{TM}$647 fluorophores were excited in TIRF mode (penetration depth ~200 nm) using a 640 nm laser with a total output power of 70 mW (Toptica iCHROME MLE multi laser engine), resulting in a peak excitation intensity of ~2 kW per cm$^2$. The exposure time was set to 10 ms. Rhodamine dye (R18) was excited with a peak intensity of ~ 0.1 kW per cm$^2$ at 532 nm with an exposure time of 30 ms. The fluorescence signal was detected with an EMCCD camera (Andor iXON Ultra) using an EM-gain of 60 or 2 for CF$^{TM}$647 and R18 detection, respectively. To separate the excitation light from the emitted fluorescence a Quadfilter was used, combined with a long-pass filter (635LP) for the detection of CF$^{TM}$647 and a bandpass filter (570/60) for R18. For each dSTORM image 6000 frames were acquired from which the super-resolution image was reconstructed. Prior to the actual acquisition a few hundred frames were discarded during which the fluorophores were transferred to the dark state.

**Data analysis and statistics**. STED data analyses were performed with Matlab (MathWorks Inc., Massachusetts, USA) using custom-written routines to determine spot sizes and intensities. The size and intensity of individual spots were analyzed using an automated routine, as follows. Regions of interest (ROIs) within the membrane sheets were defined manually on the general membrane staining (R18 channel), along with ROIs containing only background pixels (i.e., regions on the coverslips that did not contain membrane sheets). To determine the spot positions, the membrane sheet ROIs were subjected to a median filter, to remove "salt-and-pepper" noise, and the positions of all remaining pixels (spots), whose intensities surpassed the average background intensity by 10 arbitrary units were determined. Gaussian curves were fitted onto these pixels, relying on the raw data images (not the median-filtered). If X and Y Gaussian fits suggest a true-spot, the full width at half maximum (FWHM) and integrated intensity values presented here were determined from these fits.

For dSTORM, raw images were analyzed using the localization software rapidSTORM[65] and images were reconstructed using a custom-written Matlab routine, which also allowed for drift correction by tracking fiducial markers. The reconstructed images were then further analyzed using Matlab routines for extracting the size and overall intensity (i.e., number of detection events) of the CF$^{TM}$647 spots by fitting 2D Gaussian distributions.

The spot intensities obtained from the STED images of dozens of membrane sheets where pooled together and the overall distributions were fitted. As a fit model, we use the sum of Gaussian distributions, where we force the center positions to be multiples of the center position of the single-fluorophores reference distribution

$$f(I) = \sum_{n=1}^{20} a_n \frac{1}{\sqrt{2\pi n}\sigma} e^{-\frac{(I-nI_0)^2}{2n\sigma^2}} \quad (3)$$

where $I_0$ and $\sigma$ are the center position and the width of the single-fluorophore intensity histogram, respectively (the upper limit of 20 fluorophores is safely beyond what we observe in the experiments). Thus, spots containing $n$ fluorophores would result in an intensity distribution centered around $nI_0$ with a width of $\sqrt{n}\sigma$. The amplitudes or weighting factors $a_n$ are the parameters obtained from the fit and indicate the number of spots containing $n$ fluorophores.

The fraction of spots containing $n$ fluorophores is thus

$$f_{s,n} = \frac{a_n}{\sum_{n=1}^{20} a_n} \quad (4)$$

The quantity we are interested in is the fraction of fluorophores, which are part of a spot containing $n$ fluorophores. This fraction is given by

$$f_{f,n} = \frac{na_n}{\sum_{n=1}^{20} na_n} \quad (5)$$

The two quantities $f_{s,n}$ and $f_{f,n}$ are also explained and depicted in the example shown in Supplementary Fig. 2b. Since one receptor is typically labeled with two fluorophores the amplitudes for even numbers of fluorophores are the dominant contributions. We determined the fraction of fluorophores, which label a monomer as $f_{f,1} + f_{f,2}$, i.e., every spot that contained one or two fluorophores. Accordingly, dimers of mIgM-BCRs contained three or four fluorophores, and every spot containing more than five fluorophores was considered an oligomer. From the fits we obtained $a_n$ for all amplitudes together with the respective confidence intervals, which was needed to determine the uncertainties of the monomer, dimer, and oligomer fractions using standard rules for error propagation. To determine the contribution of large clusters we first calculated the proportion from the total signal that was not monomers, dimers, and oligomers. Therefore, the rest of the signal present in the membrane sheets correspond to large clusters that were not considered true-spots due to their large size (≥700 nm FWHM) or inhomogeneous shape (bad X & Y Gaussian fits).

**Reporting summary**. Further information on experimental design is available in the Nature Research Reporting Summary linked to this article.

## Data availability
The raw data underlying Figs. 1d, g, j and 3c, f, i and Supplementary Figs. 2 and 5b, c, d are provided as a Source Data file. Custom-made Matlab codes can be obtained at fopazo@gwdg.de.

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

## Acknowledgements

We thank Silvio O. Rizzoli for help with the Matlab analysis routines and helpful advice. This work was supported by the *Deutsche Forschungsgemeinschaft* (DFG) through Cluster of Excellence Nanoscale Microscopy and Molecular Physiology of the Brain (CNMPB), the TRR130 and the grant EN834/3-1.

## Author contributions

M.A.G.d.C. designed and performed experiments, analyzed data, and wrote the manuscript, H.W. designed and performed experiments, analyzed data, and wrote the manuscript, C.H. designed and performed experiments. S.SI. designed and performed experiments and analyzed data. M.B. and M.T. provided CLL immunoglobulin sequences. N.E. designed and performed experiments, analyzed data, designed the research, and wrote the manuscript, F.O. designed and performed experiments, analyzed data, designed the research, and wrote the manuscript.

## Additional information

**Competing interests:** The authors declare no competing interests.

