## [Peer Review File · Nature Communications]

Reviewers' comments:

Reviewer #1 (Remarks to the Author):

This manuscript addresses the interesting and timely question of the oligomeric state of BCR in resting cells using STED microscopy in combination with a quantitative intensity based analysis. Overall, the spot-intensity analysis of STED data presented in this manuscript could provide improvements over current methods using localization-based data, as those data sets often suffer from additional issues of artificial clustering arising from multiple observations of single fluorophores. The authors also present a labeling strategy to control the number of fluorophores per antibody label. However, beyond these improvements, the analysis presented appears to lack the rigor required to back up the strong statements made, or if it does, important considerations are not included in the text. Along these lines, there is little discussion of errors that arise and how they propagate through the analysis. These concerns, along with the lack of a useful discussion regarding inherent limitations and how they influence results prevents this reviewer from being able to independently evaluate the conclusions drawn.

Major and minor concerns are detailed below.

1. In general, the authors mainly focus on average metrics of BCR "spots" and tend to use language that slightly over-generalizes the average properties of spots to the entire distribution. I think that this leads to some over-statements of what the results mean. For example, in the abstract, it is stated that "Our direct visualization method reveals that mIgM-BCRs that deliver tonic survival signals to mature B cells do not adopt larger oligomeric complexes within the plasma membrane of resting cells." (pg 2 line 12-14). I think this a bit of an over-statement because without a measurement of which specific receptors on the membrane are "signaling", it can't be determined which structures in the distribution of structures is responsible for tonic signaling. Even in unstimulated cells, there is a small population of oligomers and clusters (as shown in Fig 1 c-d), and it could be possible that this is the "signaling active" population.

2. Regarding the sample preparation for plasma membrane sheets - BCRs were labeled, and in some cases stimulated, for long periods of time at 4C, followed by fixation at 4C. Is there not some concern that the arrangement of BCRs on the plasma membrane could be significantly altered by prolonged incubation at cold temperatures? If the organization of BCRs is influenced by some active process, such as actin/myosin activity, presumably this activity would be very different at 4C than even room temperature. The membrane sheets preparation technique is cited, and the authors do explain that they label at cold temperatures to avoid endocytosis, but it would be helpful if they could also comment on why incubation in the cold is not an issue for their measurement.

3. The spot intensity-based measurement seems like a valid approach, but for this method to work well I would expect that the histograms of fluorescence intensity for the BCR spots should be recognizable as a superposition of Gaussians spaced at regular intervals. Looking at the plots of Gaussian fits to BCR spot intensity distributions (sup fig 2), there are clear deviations between the data and the sum of the Gaussian fits. What metrics are used to ensure that the data actually fit a superposition of Gaussians? Can these deviations be accounted for or at least described?

Also, how is the variation in size of BCR complexes taken into account in the spot intensity analysis? It is stated that any spots larger than 700nm FWHM are excluded from the analysis, but 700nm is far larger than the resolution of STED. Is there a distribution of BCR spot sizes over several hundred nm? If so, it seems like simply fitting a whole (say, 200nm) spot to a Gaussian and using the peak intensity of that fit would yield an inaccurate measurement of how many fluorophores are in the spot, because

they could presumably be spread out in 2D (i.e. not exactly super-imposed). Instead, one would need to fit some measurement of integrated intensity or else perform a fit of superimposed 2D Gaussians corresponding to the intensity/width of single fluorophores. The fitting model, as stated in the methods (pg. 23 line 19) only makes sense if all "true spots" consist of multiple BCRs that all have identical x,y positions, and it's unclear how well this assumption actually corresponds to the data.

4. I think there needs to be some accommodation for the fact that affibodies seem to label IgM at a 1.7:1 ratio instead of a 2:1 ratio. This means that there is some variation in labeling stoichiometry across the population of BCRs, so some population is labeled with less than 2 affibodies (I note that BCRs that are labeled with 0 affibodies cannot be recorded in this experiment). The authors make a point of saying that the affibodies have almost perfect 1:1 fluorophore:affibody labeling efficiency, and this is great, but if the number of fluorophores per spot is being equated with number of IgMs, then consistency of the labeling ratio of affibodies:IgM is also important. In other words, if it is crucial for the affibodies to have 1:1 stoichiometry with fluorophores, then the 2:1 stoichiometry of affibodies to IgM would also be crucial, and the variation in affibody:IgM ratio is as much a variation in labeling efficiency as variation in the fluorophore:affibody ratio would be. It's clear from the data itself that this variation exists (e.g. in Sup. Fig 2 A,C there is a significant population of spots labeled with only 1 fluorophore), and this highlights the need to take incomplete labeling into account when calculating the fractions of monomers, dimers, etc. I think it's expected that the labeling can't be perfect, but it would be helpful for the authors to acknowledge this source of error and provide some estimate of how this error leads to variation in the quantification of spot stoichiometry, etc.

Related to this, I think the quantification of the fluorescence intensity of single affibodies and the affibody:BCR labeling ratio (shown in Sup Fig 1d) requires more detail. The authors measure fluorescence intensities of single affibodies on a coverslip and pull out an average intensity for the spot intensity analysis. A similar measurement is made of single BCR/affibody conjugates to determine the average labeling ratio. I think the statistics of both of these measurements should be reported in more detail, because the success of the spot intensity analysis depends on the consistency of the intensity of single affibodies and single BCRs. Specifically, how wide is the distribution of intensities as how much variation is there relative to the average? Is this variation small enough to enable consistent results in the spot intensity analysis, and how would it propagate through the analysis? Just by looking at the images supplied in Sup. Fig. 1d, it looks like there is quite a lot of variation in intensity among the single affibodies and BCRs, so I don't think a comparison of the average values of the intensities is sufficient to characterize the labeling ratio.

A consequence of the variation in labeling is that single spots cannot be definitively assigned as monomers, dimers, etc., because a dimer, for example, could potentially be labeled by anywhere between 0 (though these would be rare) and 4 fluorophores. Because of this, I think some of the language that states definitively that certain populations of spots "are monomers" or "are dimers", etc., needs to be weakened slightly to acknowledge the error arising from the variation of labeling. This is done in some places by phrasing like "BCRs exist predominantly as monomers", but these statements ought to be qualified with an actual quantification of the variation.

5. Regarding the selection criteria for "true spots" described on pg. 7 line 22-26, what is meant by the statement "The minimum distance between individual 'true-spots' was approximately 50-60 nm, which is given by the resolution maximum of our STED instrument"? Does this mean that any spots that were not at least 50-60nm away from one another were excluded from analysis or does this mean that the other selection criteria resulted in true spots that were at least 50-60nm away?

Minor issues:

1. In the introduction and discussion sections, the authors contrast the STED technique against

"indirect visualization" methods (e.g. pg. 3 line 20). It's a little bit unclear what "indirect" means in this context. Does this refer only to proximity-detection methods such as PLA and FRET or does this also refer to other forms of diffraction-limited optical microscopy?

2. When the membrane sheets preparation procedure is described in the results, it is stated that cells are stained, fixed and then adhered to coverslips (pg. 7 lines 3-5). In the methods, it is stated that cells are stained, adhered to coverslips, and then fixed (pg. 19 lines 8-9). This should be clarified.

3. The results are compared to what would be predicted by the "activation-dissociation" model. Are the stimulation conditions used here actually comparable to experiments that support this model?

4. It is stated in the text that affibody labeling "did not induce significant activation," although a calcium response is visible in the plot shown in Supplementary material, it is just small compared to the response of Fab2 (which typically provides a robust response). What concentration of Fab2 was added? Why is the reader to conclude that this activation is not significant?

Reviewer #2 (Remarks to the Author):

Gomes de Castro¹ and colleagues demonstrate with a new imaging approach the membrane organization of a b-cell with respect to BCR where they demonstrate monomer or dimer IgM in resting B-cells. Activated B-cells form monomers. In contrast, CLL cells have greater monomer structure at baseline even in the absence of IgM stimulation due to self auto activation sequence. This is abrogated by mutating the co-localizing amino acid responsible for self recognition. Overall, this new imaging strategy calls into question some former findings of B-cell receptor membrane occupancy. There are points that can be improved.

1. The cell line created by the authors allows for testing of differential downstream BCR signaling at baseline (including BTK expression and phosphorylation) as compared to the two control lines

2. The experiments are all done derive from one IGHV un-mutated CLL patient which make the generalizability to the whole disease or even a subset of IGHV un-mutated more problematic. While the molecular biology of creating these variants is time consuming, it would add to the impact of the paper to have repeated this with several patients.

3. What are the growth features of the cells examined and are they resistant to apoptosis as compared to the parent and control line.

Reviewers' comments:

Reviewer #1 (Remarks to the Author):

This manuscript addresses the interesting and timely question of the oligomeric state of BCR in resting cells using STED microscopy in combination with a quantitative intensity based analysis. Overall, the spot-intensity analysis of STED data presented in this manuscript could provide improvements over current methods using localization-based data, as those data sets often suffer from additional issues of artificial clustering arising from multiple observations of single fluorophores. The authors also present a labeling strategy to control the number of fluorophores per antibody label. However, beyond these improvements, the analysis presented appears to lack the rigor required to back up the strong statements made, or if it does, important considerations are not included in the text. Along these lines, there is little discussion of errors that arise and how they propagate through the analysis. These concerns, along with the lack of a useful discussion regarding inherent limitations and how they influence results prevents this reviewer from being able to independently evaluate the conclusions drawn.

Major and minor concerns are detailed below.

1. In general, the authors mainly focus on average metrics of BCR "spots" and tend to use language that slightly over-generalizes the average properties of spots to the entire distribution. I think that this leads to some over-statements of what the results mean. For example, in the abstract, it is stated that "Our direct visualization method reveals that mIgM-BCRs that deliver tonic survival signals to mature B cells do not adopt larger oligomeric complexes within the plasma membrane of resting cells." (pg 2 line 12-14). I think this a bit of an over-statement because without a measurement of which specific receptors on the membrane are "signaling", it can't be determined which structures in the distribution of structures is responsible for tonic signaling. Even in unstimulated cells, there is a small population of oligomers and clusters (as shown in Fig 1 c-d), and it could be possible that this is the "signaling active" population.

We thank the reviewer for this valid comment. We have modified the sentence in question, which now reads "Our direct visualization method reveals that mIgM-BCRs that deliver tonic survival signals to mature B cells do not abundantly adopt larger oligomeric complexes within the plasma membrane of resting cells.". Furthermore, we discuss the possibility that the few apparent BCR oligomers and clusters that we find in resting cells are a signaling active population (page 15, lines 18-19).

2. Regarding the sample preparation for plasma membrane sheets - BCRs were labeled, and in some cases stimulated, for long periods of time at 4C, followed by fixation at 4C. Is there not some concern that the arrangement of BCRs on the plasma membrane could be significantly altered by prolonged incubation at cold temperatures? If the organization of BCRs is influenced by some active process, such as actin/myosin activity, presumably this activity would be very different at 4C than even room temperature. The membrane sheets preparation technique is cited, and the authors do explain that they label at cold temperatures to avoid endocytosis, but it would be helpful if they could also comment on why incubation in the cold is not an issue for their measurement.

We performed labelling of BCRs at ice-cold temperatures mainly for two reasons: first, to prevent membrane trafficking (Punnonen et. al. (1998) Eur J Cell Biol 75:344–352) and hence to avoid the rapid steady-state endocytosis of BCRs at 37°C (Patel & Neuberger (1993) Cell 74:939-946). Although we cannot completely rule out that low temperatures affect the distribution of membrane

proteins, our experience and the literature suggest that membrane proteins should remain in place at low temperatures since the membrane fluidity is strongly reduced. This allows for visualization of surface BCRs that otherwise (at 37°C) will be continuously trafficked/recycled (as in Figure 1 for Reviewers). Furthermore, there is evidence of only minimal alterations of the actin cytoskeleton at low temperatures (Zimmerle et al. (1986) *Biochemistry* 25:6432–6438). Therefore, we consider that doing this “snap-shot” of BCRs on the B cell surface is performed best under ice-cold conditions. Nevertheless, we tested staining of BCRs at 37°C and consistent with the literature already after 5 minutes we see that IgM-BCRs get endocytosed (see Figure 1 for Reviewers).

The second reason for BCR staining at low temperatures is owed to the observations that stimulation of BCRs at physiological temperatures causes their rapid internalization within minutes (Drake et al. (1989) *J Immunol* 143:1768-1776, Pure & Tardelli (1992) *PNAS* 89:114-117, Hou et al (2006) *PLOS Biol* 4(7) e200). Accordingly, when we made the BCR stimulation experiment using a divalent F(ab')₂ at 37°C, we detected way less BCRs in our membrane sheets many of which were totally devoid of fluorescence signals, suggesting that BCRs got rapidly internalized and we were thus unable to detect them. In line with this, it is also important to keep in mind that aldehyde fixation is not an instantaneous process, which means that even if we add the fixative reagents soon after the stimulating F(ab')₂ there is probably plenty of time for internalization of BCRs at 37°C.

Therefore, in order to reduce the movement and trafficking of BCRs as effectively as possible in our experiments, fixation was always performed at 4°C. Furthermore, evidence from the literature suggests that ice-cold fixation minimizes artifacts like cellular swelling and fixation-induced blebbing of membranes (Zhao, S. et al. *FEBS Open Bio* 4, 190–199 (2014), Richter, K. N. et al. *EMBO J* 37, 139–159 (2018).

We now made this more explicit in our manuscript (page 7, lines 12-13 and page 13 lines 24-25).

Figure 1 for Reviewers. Live staining using affibody-635P on wild type RAMOS cells. Cells were stained at 37°C for only 5 minutes or on ice for 30 minutes. Additionally, a set of cells was further stimulated with a polyclonal F(ab')₂. White arrow heads indicate the internalization of BCR-affibody-635P complex. Z-stack of confocal images are shown for every condition. Scale bars represent 5 μm.

3. The spot intensity-based measurement seems like a valid approach, but for this method to work well I would expect that the histograms of fluorescence intensity for the BCR spots should be recognizable as a superposition of Gaussians spaced at regular intervals. Looking at the plots of Gaussian fits to BCR spot intensity distributions (sup fig 2), there are clear deviations between the data and the sum of the Gaussian fits. What metrics are used to ensure that the data actually fit a superposition of Gaussians? Can these deviations be accounted for or at least described?

Also, how is the variation in size of BCR complexes taken into account in the spot intensity analysis? It is stated that any spots larger than 700nm FWHM are excluded from the analysis, but 700nm is far larger than the resolution of STED. Is there a distribution of BCR spot sizes over several hundred nm? If so, it seems like simply fitting a whole (say, 200nm) spot to a Gaussian and using the peak intensity of that fit would yield an inaccurate measurement of how many fluorophores are in the spot, because they could presumably be spread out in 2D (i.e. not exactly super-imposed). Instead, one would need to fit some measurement of integrated intensity or else

perform a fit of superimposed 2D Gaussians corresponding to the intensity/width of single fluorophores. The fitting model, as stated in the methods (pg. 23 line 19) only makes sense if all "true spots" consist of multiple BCRS that all have identical x,y positions, and it's unclear how well this assumption actually corresponds to the data.

The fact that the individual Gaussian contributions are not clearly recognizable in the histograms by eye is partially due to the width and the mean of the single-fluorophore-intensity-distribution (SFID) being on the same order of magnitude. There are intrinsic mechanisms causing a broad distribution of intensities as e.g. the random distribution of the fluorophores' dipole orientations. Variations in the microenvironment seen by the fluorophores might potentially be an additional source of broadening of the SFID. As a consequence, the individual Gaussian contributions overlap significantly and, in particular with experimental noise on top, are not easily recognizable. However, we can still extract information from the histograms by using our knowledge about the measured SFID. We agree that without using this knowledge the fit results would not be robust at all. In our approach where we determine only the amplitudes of the Gaussian distribution from the fit we obtain robust results with reasonably small confidence intervals (extracted from the built-in Matlab "fit"-function). Nevertheless, we agree that the data do not perfectly fit the model. While part of the deviations can be attributed to Poissonian number fluctuations (with e.g. 25 occurrences in one histogram bin the observed number intrinsically scatters by +/-5 events), there are some deviations that do not look purely statistical. We think that slight deviations might potentially be due to instrument drift during one measurement day. Although we carefully calibrate the STED microscope before starting a measurement, we cannot fully exclude that during the course of a measurement the amount of light detected per fluorophore changes due to slight misalignment of the instrument. In addition, variations in the microenvironment of the fluorophores might slightly alter the emitted intensities. Despite these potential sources of uncertainty, we still think that our model is the most reasonable for describing the expected intensity histogram because each spot must consist of an integer number of fluorophores and we do not use any further assumptions in the model besides the measured single-fluorophore-intensity-distribution. We now acknowledge some of the limitations of our model in the discussion section (page 15, lines 5-10).

To emphasize the suitability of the model we agree that it is a good idea to specify metrics for the quality of the fits and thus added the R^2 -values (typically 0.8-0.9) directly to the graphs (Supp. Figs. 2 and 5).

In the methods part we added some information on the propagation of errors. In short, from the fit we obtain confidence intervals for each parameter (i.e. for each amplitude), and we use standard error propagation rules to obtain the errors for the monomer, dimer and oligomer fractions.

Concerning the distribution of sizes: we attach a graph with the size histograms showing indeed differences between the samples. It is, however, difficult to extract quantitative information from these histograms, because the observed spot size, in contrast to intensity, is not expected to grow linearly with the amount of fluorophores. To obtain the spot intensities we fit the spots with 2D Gaussian functions of variable sizes and we do indeed extract the integrated intensities for the histograms, i.e. exactly the quantity that reflects the amount of fluorophores. We clarified that also in the manuscript by adding the sentence "For further analysis we use the total intensity (in the following "intensity" always means "total intensity") of each spot to infer the number of fluorophores". Fitting superimposed 2D Gaussians would not help because the expected distance of 2 fluorophores is well below our resolution limit, and the fit would also be far less robust, because we do not have a priori knowledge about the positions and the number of fluorophores in one spot. In contrast, fitting a single 2D Gaussian yields very reliable results, in particular for the integrated intensity. Even if the "true shape" of the intensity distribution deviates slightly from a 2D Gaussian, the integrated intensity can still be determined very precisely

4. I think there needs to be some accommodation for the fact that affibodies seem to label IgM at a 1.7:1 ratio instead of a 2:1 ratio. This means that there is some variation in labeling stoichiometry across the population of BCRs, so some population is labeled with less than 2 affibodies (I note that BCRs that are labeled with 0 affibodies cannot be recorded in this experiment). The authors make a point of saying that the affibodies have almost perfect 1:1 fluorophore:affibody labeling efficiency, and this is great, but if the number of fluorophores per spot is being equated with number of IgMs, then consistency of the labeling ratio of affibodies:IgM is also important. In other words, if it is crucial for the affibodies to have 1:1 stoichiometry with fluorophores, then the 2:1 stoichiometry of affibodies to IgM would also be crucial, and the variation in affibody:IgM ratio is as much a variation in labeling efficiency as variation in the fluorophore:affibody ratio would be.

It's clear from the data itself that this variation exists (e.g. in Sup. Fig 2 A,C there is a significant population of spots labeled with only 1 fluorophore), and this highlights the need to take incomplete labeling into account when calculating the fractions of monomers, dimers, etc. I think it's expected that the labeling can't be perfect, but it would be helpful for the authors to acknowledge this source of error and provide some estimate of how this error leads to variation in the quantification of spot stoichiometry, etc.

We are aware of the importance to properly estimate the stoichiometry of the affibody:IgM complexes. A ratio of 2:1 (affibody:IgM) is at least theoretically expected, since affibodies are modified versions of Staphylococcus Protein A, which binds the Fc domain of immunoglobulins in a 2:1 ratio (Deisenhofer, Biochemistry, 20(9):2361-70 (1981)). In the original manuscript, we tested the complex stoichiometry using a monomeric IgM molecule and obtained a ratio of 1.7. We considered this experimental value to be a rather good approximation, considering that two technical problems were playing against the theoretical value of 2.0. First, the affibody, like every affinity probe, has a binding K_d , which unfortunately is not provided by the commercial supplier. Therefore, we experimentally determined the amount of affibody needed to achieve saturating staining of BCRs on live cells (Supplementary Fig. 1). However, even at saturating conditions, some of the target molecules will not be fully labeled (as suggested from all the intensity distribution profiles). Whether this is due to the K_d of the affibody or sterical issues of the binding partners remains unknown. Second, also in our in vitro affibody:IgM binding experiments we used an excess of affibody in relation to monomeric IgM, to achieve saturated binding. However, this approach was less than perfect, since it inevitably left a proportion of free single affibodies that bound to the glass coverslips (next to the IgM-bound affibodies). Detection of those free single affibodies most likely contributed in reducing the theoretical ratio of 2.0.

During the review process we performed two new approaches to improve our in vitro binding experiment. In the first approach, as before, we co-incubated an excess of fluorescent affibody this time with a recombinant monomeric IgM-Fc domain (monomeric means two human μ heavy chain constant domains per molecule). Then, to remove free affibody molecules, we passed the mixture through agarose beads coupled to IgM molecules, allowing the excess free affibody molecules to bind to the immobilized IgM. The cleared mixture should be (more or less) free of unbound affibody. Using this newly improved experimental protocol, the resulting ratio of affibody to IgM-Fc domain is 1.85 (as displayed in the updated Supplementary Fig. 1). The second approach involved microscale thermophoresis (Kazemier et. al. (2017) Nucleic Acids Res 45, 5913–5919) to determine the affinity of the affibody for the IgM-Fc domain and then the stoichiometry of the complex. The results of these new measurements suggest a 2:1 (affibody:IgM-Fc) binding stoichiometry. We included this additional information in the updated Supplementary Fig. 1.

Related to this, I think the quantification of the fluorescence intensity of single affibodies and the affibody:BCR labeling ratio (shown in Sup Fig 1d) requires more detail. The authors measure fluorescence intensities of single affibodies on a coverslip and pull out an average intensity for the spot intensity analysis. A similar measurement is made of single BCR/affibody conjugates to determine the average labeling ratio. I think the statistics of both of these measurement should be reported in more detail, because the success of the spot intensity analysis depends on the consistency of the intensity of single affibodies and single BCRs. Specifically, how wide is the distribution of intensities as how much variation is there relative to the average? Is this variation small enough to enable consistent results in the spot intensity analysis, and how would it propagate through the analysis? Just by looking at the images supplied in Sup. Fig. 1d, it looks like there is quite a lot of variation in intensity among the single affibodies and BCRs, so I don't think a comparison of the average values of the intensities is sufficient to characterize the labeling ratio.

We would like to thank the reviewer for this well-founded comment. We now included the distribution of fluorescence intensities of the improved in vitro binding experiment described above in the updated Supplementary Fig. 1. The fluorescence intensity distribution of single Star635P-labeled affibody molecules is rather narrow. However, the distribution profile of the affibody:IgM-Fc complexes is slightly broader, which is most likely due to some “contamination” by single unbound affibodies on the glass coverslips and/or incomplete labelling of some IgM-Fc domains (by only one affibody molecule). From these results, it is obvious that our methodology has its limitations when it comes to the exact quantification of larger IgM clusters, since the binding of our fluorescent affibodies to their target is not perfectly linear. For this reason, we categorized every spot in plasma membrane sheets that is brighter than the equivalent of 5 single fluorophores/affibodies as ‘clusters’ of BCRs, since we cannot determine the number of IgM molecules in these structures with satisfactory precision. However, we did not see many of these brighter clusters. Most of our analyzed spots displayed a fluorescence intensity equivalent to one and up to four single fluorophores/affibodies (Supplementary Fig. 2). This suggests that most BCRs in the plasma membrane exist as monomers, dimers or perhaps even trimers (e.g. if three IgM molecules are recognized by only one affibody molecule each). Importantly, we did not detect significant numbers of BCRs that would be consistent with an organization in large protein islands in the absence of a cross-linking reagent as previously suggested.

A consequence of the variation in labeling is that single spots cannot be definitively assigned as monomers, dimers, etc., because a dimer, for example, could potentially be labeled by anywhere between 0 (though these would be rare) and 4 fluorophores. Because of this, I think some of the language that states definitively that certain populations of spots “are monomers” or “are dimers”, etc., needs to be weakened slightly to acknowledge the error arising from the variation of labeling. This is done in some places by phrasing like “BCRs exist predominantly as monomers”, but these statements ought to be qualified with an actual quantification of the variation.

We agree with the reviewer that despite using saturating amounts of fluorophore-conjugated affibody for BCR labelling there is a certain uncertainty regarding the exact stoichiometry of IgM-BCRs on the cell surface. To acknowledge this uncertainty in our definition of BCR monomers, dimers, etc., we now clearly state that we define apparent monomeric, dimeric or oligomeric organization of IgM-BCRs depending of their fluorescent spot intensity profiles (page 8, lines 20-21). Furthermore, we use phrasings like “fluorescent spots that are consistent with BCR monomers” wherever possible.

5. Regarding the selection criteria for “true spots” described on pg. 7 line 22-26, what is meant by

the statement “The minimum distance between individual ‘true-spots’ was approximately 50-60 nm, which is given by the resolution maximum of our STED instrument”? Does this mean that any spots that were not at least 50-60nm away from one another were excluded from analysis or does this mean that the other selection criteria resulted in true spots that were at least 50-60nm away?

We agree that this sentence is less than optimal and might be misleading. There are no selection criteria concerning the minimum distance of two spots, but due to the resolution limit of our microscope it is not possible to resolve any two spots closer than 50-60nm. We clarified this in the manuscript by rephrasing the sentence to “Due to the resolution limit of our STED instrument it is not possible to resolve any two spots closer than 50-60nm” (page 8, lines 9-12).

Minor issues:

1. In the introduction and discussion sections, the authors contrast the STED technique against "indirect visualization" methods (e.g. pg. 3 line 20). It's a little bit unclear what "indirect" means in this context. Does this refer only to proximity-detection methods such as PLA and FRET or does this also refer to other forms of diffraction-limited optical microscopy?

Indeed, we intended to refer to assays like bimolecular fluorescence complementation (BiFC) or the proximity ligation assay (PLA). We clarified this point in the introduction (page 3, lines 20-21 and page 14, lines 1-7). We kept in the Introduction and discussion the references of the works we refer to with these “indirect methods”.

2. When the membrane sheets preparation procedure is described in the results, it is stated that cells are stained, fixed and then adhered to coverslips (pg. 7 lines 3-5). In the methods, it is stated that cells are stained, adhered to coverslips, and then fixed (pg. 19 lines 8-9). This should be clarified.

We apologize for this inconsistency and would like to thank the reviewer for noticing it. We now made clear throughout the different sections of the manuscript that the cells were first stained, then washed, then fixed and then settled to coverslips.

3. The results are compared to what would be predicted by the "activation-dissociation" model. Are the stimulation conditions used here actually comparable to experiments that support this model?

We have used conventional established BCR stimulation conditions, i.e. polyclonal bivalent F(ab')₂ fragments to human IgM. This type of reagent has been used for decades throughout the B cell community to activate B cells via their BCRs. The dissociation activation model is not restricted to a certain type of stimulating agent (such as antigen) and hence does not exclude stimulating agents like F(ab')₂ fragments. Rather the dissociation activation model assumes that BCR monomers are tightly packed into large clusters (estimated to contain 30 or more monomers) in the plasma membrane of resting cells (i.e. in the absence of antigen) that would then open or dissociate after binding to a stimulating agent. Since we do see large BCR clusters only after but not before addition of a stimulating agent, our results reinforce the traditional cross-linking model of BCR activation.

4. It is stated in the text that affibody labeling “did not induce significant activation,” although a calcium response is visible in the plot shown in Supplementary material, it is just small compared to the response of Fab2 (which typically provides a robust response). What concentration of Fab2 was added? Why is the reader to conclude that this activation is not significant?

The usage of the term “not significant” is probably not optimal, since it may be interpreted in various ways. The reviewer is right in that the Ca^{2+} signal that is evoked by the affibody (used at the same concentration we have used it during the rest of the experiments) is very small compared to what is induced by the F(ab')_2 fragments. As a matter of fact, this is what we wanted to express here. The question is why is there a small Ca^{2+} signal at all when we treat the cells with the affibody? The affibody is provided by the manufacturer as a dimer that is formed by a disulphide bond between the single cysteine residues at the C-terminus of the affibody molecules. This disulphide bond is reduced with 100 mM of TCEP before the conjugation of a fluorophore. However, even though we use a large molar excess TCEP and later of fluorophore over affibody in the conjugation reaction, we cannot completely exclude the re-assembly (oxidation) of a few affibodies forming dimers via disulfide bonds. This small fraction of dimeric (non-fluorescent) affibody is most likely responsible of a minimal clustering of BCRs and thus also of the small Ca^{2+} signal that we see. Due to the very low molecular weight of the affibody (~7kDa), we cannot efficiently separate the traces of ~14kDa dimers, since the MW difference between monomeric and dimeric affibody is simply too small to allow for a size exclusion chromatography or dialysis. We have addressed these issues in the manuscript (page 6, lines 6-9) and have replaced the wording “not significant” with “not substantial”.

Reviewer #2 (Remarks to the Author):

Gomes de Castro and colleagues demonstrate with a new imaging approach the membrane organization of a b-cell with respect to BCR where they demonstrate monomer or dimer IgM in resting B-cells. Activated B-cells form monomers. In contrast, CLL cells have greater monomer structure at baseline even in the absence of IgM stimulation due to self auto activation sequence. This is abrogated by mutating the co-localizing amino acid responsible for self recognition. Overall, this new imaging strategy calls into question some former findings of B-cell receptor membrane occupancy. There are points that can be improved.

1. *The cell line created by the authors allows for testing of differential downstream BCR signaling at baseline (including BTK expression and phosphorylation) as compared to the two control lines*

We have analyzed baseline signaling of the CLL-derived BCRs in our transfected DG75 cells by testing global tyrosine phosphorylation following induction of the BCR variants with Doxycycline. However, we could not observe clear signs of BCR-autonomous tyrosine phosphorylation events under these experimental conditions.

Increased basal phosphorylation of BCR pathway components has been observed before in CLL samples, however, the reports vary considerably in the extents to which intracellular signaling proteins were phosphorylated. Whereas some papers reported a strong constitutive global tyrosine phosphorylation in CLL cells (Contri et al., 2005, J Clin Invest 115(2):369-378), others observed a specific phosphorylation of only a few signaling molecules such as the Erk MAP kinase in a fraction of CLL samples, whereas e.g. phosphorylation of Akt was not observed (Muzio et al., 2008, Blood 112(1):188-195). In contrast, in a murine model of CLL,

phosphorylation of Akt was observed together with heightened basal phosphorylation of Btk (Singh et al., 2017, Oncotarget 8(42):71981-71995). Another study found that Btk is constitutively phosphorylated only in a fraction of CLL samples (Herman et al., 2011, Blood 117(23):6287-6296). Also substantial baseline phosphorylation of the central BCR transducer kinase Syk was found only in a fraction of the CLL samples tested (Gobessi et al., 2009, Leukemia (23(4):686-697).

Together, the ability of CLL-derived BCRs to transduce a constitutively elevated baseline signal appears to be highly variable. The capacity of CLL-BCRs to do so may also be influenced by the (over-) expression of co-factors such as ZAP-70, Lyn or others.

The CLL-BCR that we have tested here may belong to the fraction that has only a limited capacity to induce constitutive activation of BCR signaling pathways. Alternatively, the cellular background of the DG75 Burkitt lymphoma cells may not be adequate to study signaling of CLL cells. Nevertheless, the organization of the tested CLL-BCR on the plasma membrane clearly differed from BCRs in wild-type and Burkitt lymphoma B cells and as such this CLL-BCR proves that our methodology has the potential to reveal even small differences in the topology of cell surface proteins. It would be very appealing to investigate the plasma membrane organization of CLL-derived BCRs that have a previously characterized autonomous signaling capacity. Unfortunately, this endeavor turns out to be beyond the scope of this study (see below), but will be interesting to pursue in the future.

2. The experiments are all done derive from one IGHV un-mutated CLL patient which make the generalizability to the whole disease or even a subset of IGHV un-mutated more problematic. While the molecular biology of creating these variants is time consuming, it would add to the impact of the paper to have repeated this with several patients.

We found this suggestion very appealing and thus we have tried our best to analyze more CLL-derived BCRs. We have generated four additional sets of expression vectors for CLL-derived Ig-heavy and Ig-light chains and expressed them as before. However, to our disappointment these newly generated CLL-derived IgM-BCRs were expressed at the cell surface only in low amounts as compared to the original CLL-BCR and as compared to BCRs of primary and Burkitt lymphoma B cells. As a result, analysis of these additional CLL-BCRs was very difficult and time consuming. In summary, in line with our previous observations, the data are indicating an enhanced oligomerization of the new CLL-BCRs. However, due to the low IgM surface expression the quality of the data does not meet our standards. Hence, we have decided to not include them in our manuscript. Searching for solutions to optimize surface expression of the new CLL-BCR constructs has been time consuming and was not satisfactory so far. Maybe it is an intrinsic feature of the CLL-BCRs that their surface expression is low and we may not be able to improve it at all. So, much to our regret we cannot present data on additional CLL-BCRs at this point.

3. What are the growth features of the cells examined and are they resistant to apoptosis as compared to the parent and control line.

We have analyzed the growth features of our DG75 cell following induction of the CLL-BCR variants with Doxycycline. The results given Figure 2 for Reviewers show that Doxycycline treatment slowed down cell growth to some extent. However, this effect was also seen in the control cells lacking expression of a BCR. Hence, the expression of this particular CLL-BCR has no obvious effect on cell growth.

Figure 2 for Reviewers: The same DG75 transfectants as in manuscript Fig. 3 were cultured in the absence (blue curves) or presence (red curves) of 5 μ M Doxycycline for four days. All cells were seeded at a density of 0.2×10^6 cells per ml on day 1. Subsequently, the concentrations of the cells were measured every 24 hours and were normalized to the seeding concentration.

We have also tested whether induction of the CLL-BCR induces apoptosis in the cells by staining with AnnexinV and 7-AAD (see Figure 3 for Reviewers). In line with effect of Doxycycline on cell growth, we observed a clear shift in AnnexinV staining in all cells (including BCR-negative cells). As before, there was no evidence for CLL-BCR-autonomous apoptosis signaling in the cells.

Figure 3 for Reviewers. DG75 cells used in the manuscript Fig. 3 were cultured in the absence or presence of 5 μ M Doxycycline to induce expression of CLL-derived IgM-BCRs for 24 and 48 hours (see also Supplementary Fig. 5a). Subsequently, cells were stained with BV421-conjugated AnnexinV (eBiosciences) and 7-AAD to label early and late apoptotic cells, respectively.

Reviewer 3 (Remarks to the Author):

This is a very well written report about the organization of B-cell receptor/BCR on the surface of normal B cells and CLL lymphocytes, and the resting conditions and after stimulation, using a super-resolution microscopy approach. The authors report that normal B cells contain BCRs as monomers and dimers, forming more complex clusters upon stimulation. Contrast, CLL cells already form dimers and oligomers at rest due to auto-aggregation. These findings confirm the concept that BCR aggregation is a first step up on BCR activation, which can occur spontaneously in CLL due to auto-aggregation. The manuscript is very well written and highly relevant for normal B cells biology, but also for CLL pathophysiology. The data are well presented and discussed.

My only concern is related to the fact that the authors only studied CLL BCRs from patients with unmutated IgHV. Patients with mutated versus unmutated IgHV are clinically behaving very differently, presumably due to great differences in signaling capacity and dependency upon BCR

signaling. It remains controversial whether BCR auto aggregation is a phenomenon that exists in both, mutated and unmutated CLL cases. The office may want to consider, or may already have done experiments to compare spontaneous BCR aggregation in cases with unmutated versus mutated CLL. If not yet studied, such experiments would greatly increase the impact of this work related to CLL. If not feasible, the data still remain highly interesting and of high importance.

We would like to thank the reviewer for the positive opinion on our work and the valid advice to also study mutated CLL-BCRs. As already outlined in the response to Reviewer 2, we have cloned and expressed four additional CLL-BCRs, one of which was derived from a mutated CLL sample. However, even after several attempts to optimize expression levels of these additional CLL-derived BCRs, the quality of the imaging data was not satisfactory. We keep on searching for a solution to this problem, however, it means that we cannot provide information on the organization of mutated CLL-BCRs at this stage.

Reviewers' comments:

Reviewer #1 (Remarks to the Author):

I am grateful for the clarification on some of our questions in the rebuttal statement, but am disappointed that the intention behind some of our previous critique did not translate into changes made to the revised manuscript, namely that the authors needed to more clearly state the limitations of the results presented in the manuscript (not just in the rebuttal). I do not think that new experiments are needed to clarify these issues. I do think that text similar to what was included in the rebuttal should be included in the main text of the manuscript so that the reader can put results into the proper context. Some examples (not exhaustive) are noted below.

1. One page 4 of the rebuttal, there is a long discussion regarding why the individual Gaussians are not clearly recognizable in the histograms. This seems to be a valuable discussion to include in the manuscript itself rather than in only in the rebuttal statement.

2. The authors now add the word 'apparent' before discussing numbers of monomers, dimers, and oligomers (page 8 ~line 21). This weaker language is not that useful since it is not explained what 'apparent' means. This should be done at the first occurrence of this language if it is to be included. Also, I imagine it is possible to put error bounds on these values via a statistical analysis that takes into account the incomplete labeling of BCR beyond simply stating that these variations exist. This source of error compounds with the errors arising from fitting discussed in the current manuscript. This would make this section less misleading and would strengthen the results reported. Also, the numbers given in the text are large ranges and do not reflect the actual values plotted in fig 1. It seems more appropriate to have the values match in both places they are presented.

3. I understand that 4C sample preparation conditions are important for maintaining BCR at the cell surface, but it should be clearly stated early in results and there should be a discussion of how this might impact the interpretation of results. For example, from reading the manuscript, it is clear that the fixation is done on ice, but it is not obvious if the stimulation (e.g. fig 1h) is done on ice or at another temperature. Based on the text in the rebuttal, it seems like it was done on ice. If it is important to compare the stimulated condition to unstimulated or CLL conditions, then a discussion of how stimulation at 4C might impact the arrangement of BCR and its signaling seems important.

4. In the rebuttal it is stated that the weak (relative to fab2) calcium response was likely a result of a small population of unlabeled antibody dimers. This seems like a limitation that needs to be better highlighted in the main text. From Sub fig 1 c, it is clear that some cells are getting activated, even if the particular probes visualizes might not be doing the activating.

As a minor point, I recommend using the phrase 'integrated intensity' rather than 'total intensity' since it is not clear to me what total refers to.

Reviewer #2 (Remarks to the Author):

The authors have responded acceptably to my comments

Reviewer #3 (Remarks to the Author):

Thank-you for commenting, I have no further questions or concerns.

As before, we would like to thank all reviewers for taking their time to re-examine our manuscript. We were pleased to see that two reviewers are convinced by our work. We particularly thank reviewer #1 for her/his thoughtful comments, since this input gave us the chance to improve our manuscript and clarify a few aspects. Specifically, we have increased our efforts to honestly make the reader aware of the potential limitations of our methodology (pages 8, 9, 13, 15 and 16). Nonetheless, we are convinced of the reliability of the experimental approach and stand by our conclusions. Furthermore, we have added new experimental data comparing BCR stimulation at 30°C and ice-cold temperatures (page 9 and new Suppl. Fig. 1d). Finally, we have included a paragraph discussing the robustness of our method and why individual Gaussians cannot easily be recognized by eye from the raw data (pages 15-16). We hope that we have adequately addressed all points of concern in our response letter and in the revised manuscript. For your convenience, all new changes to the manuscript text are highlighted in blue.

Reviewers' comments:

Reviewer #1 (Remarks to the Author):

I am grateful for the clarification on some of our questions in the rebuttal statement, but am disappointed that the intention behind some of our previous critique did not translate into changes made to the revised manuscript, namely that the authors needed to more clearly state the limitations of the results presented in the manuscript (not just in the rebuttal). I do not think that new experiments are needed to clarify these issues. I do think that text similar to what was included in the rebuttal should be included in the main text of the manuscript so that the reader can put results into the proper context. Some examples (not exhaustive) are noted below.

1. One page 4 of the rebuttal, there is a long discussion regarding why the individual Gaussians are not clearly recognizable in the histograms. This seems to be a valuable discussion to include in the manuscript itself rather than in only in the rebuttal statement.

We agree with the reviewer's opinion and have now extended the discussion as to why individual Gaussians in the histograms are not easily recognizable by eye (page 15-16).

2. The authors now add the word 'apparent' before discussing numbers of monomers, dimers, and oligomers (page 8 ~line 21). This weaker language is not that useful since it is not explained what 'apparent' means. This should be done at the first occurrence of this language if it is to be included. Also, I imagine it is possible to put error bounds on these values via a statistical analysis that takes into account the incomplete labeling of BCR beyond simply stating that these variations exist. This source of error compounds with the errors arising from fitting discussed in the current manuscript. This would make this section less misleading and would strengthen the results reported. Also, the numbers given in the text are large ranges and do not reflect the actual values plotted in fig 1. It seems more appropriate to have the values match in both places they are presented.

We have edited the text according to the reviewer's suggestions and explain our definition of 'apparent' monomers, dimers, and so on at the first occurrence on page 8. In addition, we now specify precisely the mean percentages and standard deviations of the different mIgM-BCR arrangements (pages 8 and page 12).

3. I understand that 4C sample preparation conditions are important for maintaining BCR at the cell surface, but it should be clearly stated early in results and there should be a discussion of how this might impact the interpretation of results. For example, from reading the manuscript, it is clear that the fixation is done on ice, but it is not obvious if the stimulation (e.g. fig 1h) is done on ice or at another temperature. Based on the text in the rebuttal, it seems like it was done on ice. If it is important to compare the stimulated condition to unstimulated or CLL conditions, then a discussion of how stimulation at 4C might impact the arrangement of BCR and its signaling seems important.

We originally explained our BCR stimulation protocol in the Methods section under 'Induction of mlgM clustering'. In the revised version, we have now included a brief description also in the Results section (page 9) and in the legend to Figure 1h.

To assess the impact of cold temperatures on BCR signaling, we have tested BCR-proximal signaling reactions exemplified by Ca^{2+} mobilization at ice-cold temperatures and 30°C. Ca^{2+} mobilization on ice is slowed down and 'stretched' to some extent, i.e. both the peak and the decline of the Ca^{2+} kinetics are delayed. However, the amplitude and the overall pattern of the kinetics is similar to that at 30°C. We thus believe that it is fair to conclude that stimulation on ice is a suitable way to induce clustering of BCRs while preventing their rapid endocytosis. We describe this experiment on page 9 of the revised manuscript and show it as new Supplementary Figure 1d.

4. In the rebuttal it is stated that the weak (relative to fab2) calcium response was likely a result of a small population of unlabeled affibody dimers. This seems like a limitation that needs to be better highlighted in the main text. From Sub fig 1 c, it is clear that some cells are getting activated, even if the particular probes visualizes might not be doing the activating.

We mentioned the presumptive formation of small amounts of dimeric (non-fluorescent) affibodies during the fluorophore conjugation reaction in the manuscript on page 6 (lines 3-6). We now pick up on this property of our reagent on page 9 and again on page 15 and discuss that – at least theoretically – a few otherwise monomeric mlgM molecules may have been forced into higher ordered structures by affibody dimers. However, we believe that this minute reagent impurity does not significantly affect the conclusions of our experiments. This can be deduced from Fig. 3 g-i, which shows that the R38A variant of the CLL-BCR, which was analyzed with the same reagent, is almost exclusively monomeric.

As a minor point, I recommend using the phrase 'integrated intensity' rather than 'total intensity' since it is not clear to me what total refers to.

We have changed this accordingly (pages 8 and 25).

Reviewer #2 (Remarks to the Author):

The authors have responded acceptably to my comments

Reviewer #3 (Remarks to the Author):

Thank-you for commenting, I have no further questions or concerns.

REVIEWERS' COMMENTS:

Reviewer #1 (Remarks to the Author):

The authors have adequately addressed the vast majority of my previous concerns. Namely, the authors have included additional qualifications of their observations/conclusions. One comment that was not addressed was the suggestion to include a quantitative statistical analysis of the contributions of imperfect labeling efficiency to error in the estimates of monomers, dimers, etc. I stand by my previous statement that the manuscript would be enhanced with this component, but as is I feel it is still a valuable contribution to the BCR literature.

REVIEWERS' COMMENTS:

Reviewer #1 (Remarks to the Author):

The authors have adequately addressed the vast majority of my previous concerns. Namely, the authors have included additional qualifications of their observations/conclusions. One comment that was not addressed was the suggestion to include a quantitative statistical analysis of the contributions of imperfect labeling efficiency to error in the estimates of monomers, dimers, etc. I stand by my previous statement that the manuscript would be enhanced with this component, but as is I feel it is still a valuable contribution to the BCR literature.

We would like to thank once again reviewer #1 for her/his assessment of the latest version of our manuscript. We agree with the reviewer that it would be interesting to estimate the error associated to the labeling efficiency of our methodology. The fluorescent labeling efficiency of IgM molecules on the cell surface is influenced by a number of chemical and physical parameters inherent to the affinity probe, such as its dissociation constant, its degree of labeling, the properties of the fluorophore, etc., but also by its ligand, i.e. the IgM-BCR. There is an (at least theoretical) chance that some affibody epitopes in the native IgM-BCR complex on the cell surface are sterically blocked and thus are not accessible for affibody binding. Such sterical hindrance may be caused by protein-protein interactions in the BCR complex or associated surface molecules or by posttranslational modifications, and would interfere with complete labeling of surface BCRs. Unfortunately, there is no way by which we could assess or quantify this factor. We have stated this uncertainty (which is not specific for the affibody but applies to every affinity probe) in the Discussion section of the manuscript. Without that information, we believe that a solid quantitative statistical estimation of the parameters that contribute to the error in determining the frequencies of monomers, dimers and oligomers of IgM-BCRs, remains impossible. Nonetheless, to minimize potential errors in estimating the quantities of BCR monomers, dimers, etc., we have taken numerous precautions. These include several experimental determinations, such as the degree of labeling of the affibody to the fluorophore, the affinity of the affibody for its ligand, the stoichiometry of binding of the affibody to free, monomeric IgM, the saturating concentration of labeled affibody for staining of IgM on cell surfaces and optimization of the fixative conditions. Furthermore, we have made calibration measurements with free fluorophore-labeled affibodies in every imaging session. We are glad that reviewer #1 appreciates these efforts and considers our work a valuable contribution to the BCR literature. The constructive criticism of this reviewer throughout the review process surely has helped to improve our work.